# Vision-Language-Action Pretraining from Large-Scale Human Videos

## Abstract

Existing Vision-Language-Action models (VLA) struggle with complex manipulation tasks requiring high dexterity and generalization, primarily due to their reliance on synthetic data with significant sim-to-real gaps or limited teleoperated demonstrations. To address this bottleneck, we propose leveraging human hands as a "manipulator template", capitalizing on the rich dexterity and scalability present in web data of human manipulation. Our approach centers on physical instruction tuning, a novel training paradigm that combines large-scale VLA pretraining from human videos, perspective spatial alignment for reasoning in a unified physical space, and post-training adaptation in physical environments. Additionally, we introduce a part-level motion tokenization method which achieves millimeter-level reconstruction accuracy to model precise hand trajectories for action learning. To support our paradigm, we develop a comprehensive data curation pipeline that integrates heterogeneous sources — including motion capture, VR, and RGB-only videos — into a large-scale dataset with millions of motion-based instructional instances. We empirically show the excellence of our model in hand motion generation and instruction following, and it also scales well with model and data sizes. Importantly, we observe the expected gains in robotic dexterous manipulation as physical instruction tuning is applied.

## 1 Introduction

The advance of ChatGPT and its successors have endowed large multimodal models (LMMs) with versatile capabilities across various domains, yet their application in robotics lags behind. Recent efforts aim to bridge this gap by adapting LMMs into Vision-Language-Action models (VLAs) [44, 8], harnessing their multimodal reasoning for robotic tasks. However, these models' generalization is severely limited by a reliance on small-scale, lab-collected teleoperated demonstrations [67, 43], which are orders of magnitude smaller than the internet-scale data used to train LMMs. Consequently, VLAs lack robustness across diverse objects and environments. This data scarcity is especially acute for dexterous manipulators, where operational complexity and high hardware costs have largely restricted VLAs to simple grippers [24, 3]. Although simulators offer a path to low-cost synthetic data [88, 109], their limited diversity and persistent sim-to-real gap have thus hindered the successful deployment of dexterous hands.

Human videos offer a promising alternative for VLA training, providing abundant real-world data with a minimal reality gap. However, prior work relying on implicit learning techniques (e.g., contrastive learning [64], masked autoencoder [78], or latent action [7]) has yielded unclear learning mechanisms and limited transfer effectiveness. These methods fall short of the dramatic performance gains achieved in LMMs, where techniques like visual instruction tuning [54] have proven remarkable success. We argue this disparity stems from a fundamental difference in data structure. LMMs benefit from isomorphic data where pretraining and downstream tasks are aligned, as visual-text understanding directly translates to multimodal reasoning tasks. In contrast, VLAs face a heterogeneous alignment challenge, grappling with significant gaps between textual/2D visual inputs and the 3D physical action space, which includes critical proprioceptive requirements. Although recent explorations into explicit human-centric representation [74] show promise, their limited scale contradicts the core vision of leveraging web-scale data — the very resource that enables LMMs' success through massive pretraining. These inspire us to ask: *Can we pretrain a dexterous VLA from large-scale human videos to explicitly imitate human actions and adapt to robot manipulators via post-training?*

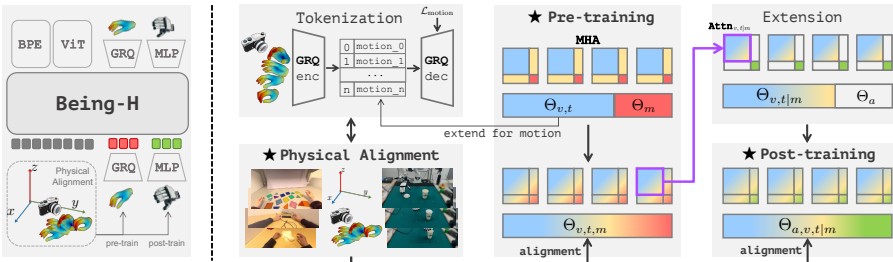

Figure 1: **Overall Being-H Pipeline(left). Physical Instruction Tuning (right): (left)** Tokenize continuous hand motions into discrete representations. Physical alignment unifies heterogeneous hand motion and robot data via coordinate alignment and transformation. **(middle)** Extend a vision-text model $\Theta_{v,t}$ with motion parameters $\Theta_m$, enabling attention for vision and text, motion, and cross modality within a unified sequence. **(right)** The model leverages pretrained cross-modal dependencies ($\Theta_{v,t|m}$), then incorporates action parameters $\Theta_a$ via post-training to produce the final VLA $\Theta_{a,v,t|m}$ for downstream tasks. The condition on $m$ in $\Theta_{v,t|m}$ indicates that the weights are influenced by pretraining on hand motions, so we denote the post-trained weights as $\Theta_{a,v,t|m}$. The green part represents action attention.

Our motivation is straightforward: the human hand is the template for dexterous manipulation [45], exhibiting unparalleled versatility in natural settings. Learning from human motion offers a powerful pathway to bridge the pretraining-downstream data heterogeneity in robotics, but scaling to massive video data presents new challenges: (1) Data Heterogeneity. Human videos span varying camera systems, coordinate frames, and recording conditions, complicating model learning, which require unifying disparate sources and embedding essential 3D spatial reasoning capabilities. (2) Hand Quantization. To preserve granular control, continuous, fine-grained movements must be discretized into language-compatible tokens without losing millimeter-level precision. (3) Robot Control Transfer. Considering morphological differences, human hand motions require careful skill transfer to ensure learned strategies adapt effectively to end-effectors.

To address these challenges, we propose an advanced yet sample-efficient dexterous VLA trained on large-scale human videos. To train this VLA, we introduce a training framework named **`Physical Instruction Tuning`** (Figure 1) by extending visual instruction tuning [53] to the physical domain via: (1) VLA pretraining on human videos, (2) perspective spatial alignment to unify heterogeneous data from diverse camera systems and recording conditions for reasoning in the physical space, and (3) post-training adaptation to ground pretrained priors in physical environments. Unlike implicit methods [7, 64], we use explicitly hand motion prior to guide robot learning. Our VLA employs a unified autoregressive architecture with shared attention for seamless cross-modal reasoning. For precise motion tokenization, we introduce a part-level tokenizer with grouped residual quantization (GRQ) [47] that achieves millimeter-level accuracy. To support large-scale learning, we curate **`UniHand`**, a comprehensive 150M dataset integrating motion capture, VR, and RGB videos across diverse manipulation tasks. To our knowledge, this is the first dexterous VLA using explicit motion modeling from large-scale human videos.

Our key contributions are: **(1) Physical Instruction Tuning.** A novel paradigm that establishes the human hand as a template prior for robot control, bridging human videos to embodied action. **(2) Part-Level Motion Tokenization.** A quantization method that achieves millimeter-level precision for continuous motions while ensuring compatibility with discrete, autoregressive models. **(3) The UniHand Dataset.** A large-scale dataset of over 150M instruction-following samples from diverse manipulation scenarios, unified via a scalable pipeline integrating motion capture, VR, and RGB videos. **(4) Being-H.** Integrating these innovations, we present the first dexterous VLA trained on large-scale videos of human-**Being H**ands. Our model enables robust cross-modal reasoning across vision, language, and motion, with effective adaptation for downstream robot tasks.

## 2 RELATED WORK

**Hand Motion Modeling.** Existing research on human hand motion primarily focuses on hand-object interaction (HOI) [35, 41] and fine-grained action precision. While current benchmarks [6, 103] capture these interactions, their reliance on motion-capture systems or multi-camera setups limits

them to tabletop scenarios. Egocentric videos [35] from head-mounted cameras offer environmental diversity but often lack precise 3D annotations. Recent progress in monocular 3D hand modeling [68, 27] now enables pseudo-label extraction from such videos, though their weak perspective assumption makes them incompatible with shifted-perspective egocentric datasets [34]. Dyn-HaMR [100] addresses this by integrating SLAM for camera tracking and occlusion-robust refinement. Beyond modeling hands in isolation, recent efforts [56] jointly focus on HOI by predicting interaction hotspots, future trajectories, and affordance. Concurrently, the field of hand-object interaction has evolved from 2D recognition/detection [32, 33, 72] to 3D motion generation. Early approaches use multi-stage pipelines [22, 13] from action labels [31, 9], while recent methods employ diffusion or autoregressive models [37] for generation, leveraging LLMs as unified backbone for long-term consistency [39]. Despite these advances, most methods overlook visual inputs until MEgoHand [110]. Inspired by these advances, we equip our model with generation capabilities pretrained on large-scale human videos to provide strong hand motion priors for downstream manipulation tasks.

**Learning VLAs from Human Videos.** The emergence of LMMs has enabled visual-language-action models (VLAs) to map perceptions to physical actions. Existing approaches [44, 8, 7] leverage massive robot data pretraining [67, 43] to improve generalization. FAST [70] improves autoregressive prediction by proposing discrete cosine transforms for fast and scalable training. Instead of using discrete action tokens, Octo [82] and RDT-1B [57] adopt a diffusion head for flexible action prediction. Despite progress, current datasets remain limited to small-scale lab collections especially for dexterous manipulators due to higher costs than grippers. While simulation offers scalability [96, 26], the sim-to-real gap persists. Human videos provide a promising alternative, with prior works extracting transferable representations like visual features [64], 3D priors [94], and interaction knowledge (e.g., affordance, contacts, and grasps) [30, 16]. Yet, these methods fail to explicitly bridge the human-to-robot action gap due to structural differences. Some approaches align visual observations through data editing (e.g., image inpainting and rendering [49], visual masking [42]), while others align action space via unified human-centric state-action representations [66, 42] or trajectory refinement using RL/physical simulation [51, 19] to produce physically plausible and smoother actions. Crucially, these efforts are limited to simple grippers, neglecting fine-grained finger motion. In this paper, we treat human hands as a universal template for the downstream manipulator. To our knowledge, this is the first to pretrain a scalable, generalizable VLA through explicit motion modeling from large-scale human videos, enabling robots to learn diverse skills from internet data.

## 3 PHYSICAL INSTRUCTION TUNING

We analyze the success factors of visual instruction tuning and propose this novel paradigm for training our dexterous VLA — Being-H, which includes three key components as shown in Figure 2

### 3.1 PRETRAINING ON HAND MOTION

We leverage human-robot anatomical similarity by pretraining on hand motion generation, treating the human hand as an ideal manipulator template and robots as simplified versions. Our pretrained VLA learns to predict MANO-parameterized motions $m = \{\theta, \mathbf{r}_{rot}, \tau, \beta\}$ (joint angles $\theta$, wrist rotation $\mathbf{r}_{rot}$, translation $\tau$, and shape $\beta$) from visual-text context. Similar to the multimodal instruction tuning in VLMs, each sample is treated as an instructional pair $\{\mathcal{X}_Q, \mathcal{X}_A\}$ and the objective is:

$$\theta^* = \arg\min_\theta \sum_{i=1}^{N} \mathcal{L}(\Theta) = -\sum_{j=1}^{L} \log P_\Theta(y_j \mid \mathcal{X}_Q, y_{1:j-1}) \tag{1}$$

where $\mathcal{X}_Q$ refers to the question inputs, $\mathcal{X}_A = \{y_{1:L}\}$ refers to target answer tokens, and $y_i$ refers to inidividual tokens in the answer sequence. Building on LMMs like InternVL3 [20], the model incorporates motion tokens into the vocabulary, enabling autoregressive generation of hand motion sequences conditioned on visual-text inputs. Due to space limitation, we introduce multimodal integration and training details in Appendix B.1 and B.3.

Inspired by [91], we treat hand movements as a foreign language by quantizing continuous motion features into discrete embeddings using a motion tokenizer during pretraining. The tokenizer encodes $T$-frame hand features $\mathcal{M} = \{m_1, m_2, ..., m_T\}$ of raw motion sequence into $\lceil T/\alpha \rceil$ $d$-dim token embeddings, where $\alpha$ denotes the temporal downsampling rate. To represent hand efficiently and

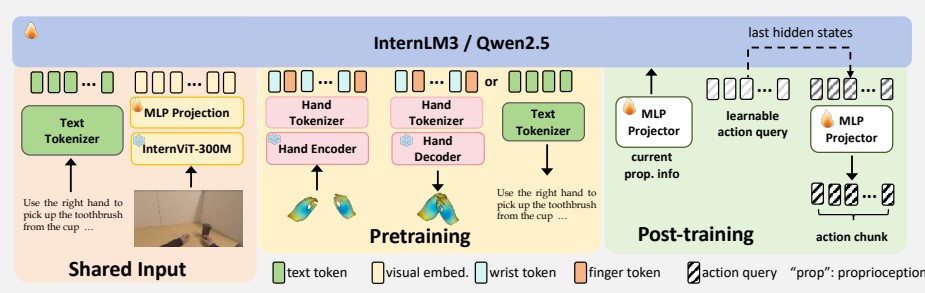

Figure 2: **Model Overview:** The text tokenizer and visual encoder are shared by both pretraining and post-training. For pretraining, a hand motion tokenizer is included for hand motion inputs and autoregressive outputs of Being-H. For post-training and downstream tasks, Being-H incorporates a set of learnable queries, whose hidden states are converted into continuous action chunks.

effectively, we use the MANO-162 as the hand motion parameterization. Each frame is encoded as $m \in \mathbb{R}^{162}$, including joint pose $\theta \in \mathbb{R}^{15 \times 6}$, global rotation $\mathbf{r}_{rot} \in \mathbb{R}^6$, and translation $\tau \in \mathbb{R}^3$. Both $\theta$ and $\mathbf{r}_{rot}$ are in the axis–angle form. The LMM vocabulary is extended by integrating $K$ discrete motion codes and special tokens `<MOT>` and `</MOT>` to mark motion block boundaries.

**Hand Motion Tokenization.** The precision of the motion tokenizer critically impacts both the quality of generated hand motions and the transferability of learned priors to downstream tasks. We therefore design a dedicated tokenizer based on GRQ [95] for expressive motion representation (Figure 3). Given a motion sequence $\mathcal{M} \in \mathbb{R}^{T \times D}$, an encoder converts it into a feature map $z \in \mathbb{R}^{\lceil T/\alpha \rceil \times d}$, which is discretized via a multi-stage residual quantization. First, the channel dimension $d$ is partitioned into $n$ groups. Each group feature $z^{(g)} \in \mathbb{R}^{\lceil T/\alpha \rceil \times d/n}$ is quantized independently using an $L$-layer residual vector quantizer (RVQ) with codebook $\mathcal{C}^{(g)}$. Each feature vector $z_i^{(g)} \in \mathbb{R}^{d/n}$ in group $g$ is quantized as:

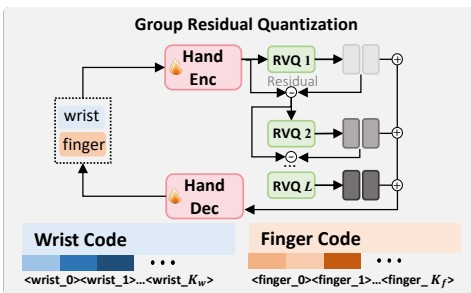

Figure 3: Architecture of part-level hand motion tokenization based on GRQ [95].

$$r_0 = z_i^{(g)}, \quad q_l = \arg\min_{c \in \mathcal{C}^{(g)}} \|r_{l-1} - c\|_2, \quad r_l = r_{l-1} - q_l \tag{2}$$

where $q_l$ and $r_l$ denote the selected code and residual at layer $l$. The final quantized representation is: $\hat{z}_i^{(g)} = \sum_{l=1}^{L} q_l$. We observe that reconstructing wrist parameters $\mathbf{r}_{rot}$ and $\tau$ is challenging due to their broad 3D distribution, despite their importance for precision. To address this, we introduce a wrist-specific loss term: $\mathcal{L}_{wrist} = \|\mathbf{w} - \hat{\mathbf{w}}\|_2^2$ where $\mathbf{w} = [\mathbf{r}_{rot}, \tau]$ and $\hat{\mathbf{w}}$ denotes the reconstructed wrist parameters. Combined with the reconstruction loss and commitment loss from VQ-VAE [86], the final objective is:

$$\mathcal{L} = \mathcal{L}_{recon} + \lambda_1 \mathcal{L}_{commit} + \lambda_2 \mathcal{L}_{wrist} \tag{3}$$

where $\lambda_{\{1,2\}}$ denotes balancing weights. We employ an exponential moving average (EMA) update strategy for the codebook. Given the higher complexity of wrist motions, we employ separate tokenizers for wrist $\{\mathbf{r}_{rot}, \tau\}$ (global pose) and finger $\{\theta, \beta\}$ (fine-grained manipulation) parameters. This part-level strategy improves feature modeling, provides explicit token semantics, and enhances the LMM's capture of structured hand dynamics. When using separate tokenizers, $\mathcal{L}_{wrist}$ is omitted. We discuss motion feature choices and tokenization details in Appendix B.2 and B.3.

**Decoding Mode.** Our pretrained VLA generates both text and motion via unified next-token prediction, with motion tokens decoded to MANO parameters. We introduce three decoding modes to balance flexibility and motion validity: **(1) Free-format Mode** which allows fully flexible autoregressive sampling without any constraints and risks producing invalid motion blocks. **(2) Block-formatted Mode** which ensures structural consistency by restricting sampling to motion tokens between `<MOT>` and `</MOT>` delimiters. For evaluation, `<EOS>` is suppressed until the target number of motion blocks is generated. **(3) Soft-formatted Mode** which focuses on evaluating local motion quality

by using a soft constraint: after each motion block, we blend predicted and ground-truth MANO parameters via their mean, then re-quantize this hybrid through the motion tokenizer. This anchors the generation in a plausible neighborhood of real trajectories, providing a reliable measure of the model's ability to produce high-quality motion in the vicinity of real trajectories.

## 3.2 PERSPECTIVE SPATIAL ALIGNMENT

Diversity of human videos introduces significant variability in camera systems, complicating effective pretraining. To alleviate this, we introduce perspective spatial alignment: a unified processing toolkit that maps videos from disparate cameras into a consistent physical coordinate system. This pipeline incorporates 3D spatial reasoning and available physical attributes to enhance geometric and perceptual consistency across datasets. Below introduce two alignment strategies adopted in this paper:

**Weak-Perspective Projection Alignment.** Inconsistent camera systems across datasets cause divergent 3D projections, impairing model depth perception and 3D reasoning. To alleviate this, we align all data to a unified weak-perspective camera space, ensuring consistent 2D-to-3D mapping and uniform scaling for objects at similar depths. Given source camera intrinsics $K = \{f_x, f_y, c_x, c_y\}$ and target $K' = \{f'_x, f'_y, c'_x, c'_y\}$, we compute scale factors and translation offsets as:

$$s_x = \frac{f'_x}{f_x}, \quad s_y = \frac{f'_y}{f_y}, \quad \Delta x = c'_x - s_x \cdot c_x, \quad \Delta y = c'_y - s_y \cdot c_y. \tag{4}$$

Each pixel $(u, v)$ in the source image is transformed as $u' = s_x \cdot u + \Delta x$ and $v' = s_y \cdot v + \Delta y$, with cropping/padding to a target resolution. For severe distortion (e.g., fisheye), we normalize the field of view (FoV) to $90°$ to minimize projection artifacts.

**View-Invariant Motion Distribution Balancing.** To prevent camera bias and ensure robust 3D generalization, we introduce this strategy that augments video-motion pairs from underrepresented sources. Our strategy varies hand pose distribution without introducing camera viewpoint and position changes. Unlike image-level augmentations like random cropping or flipping, it preserves weak-perspective consistency between hand motion and visual observations, ensuring coherent 3D understanding. Our strategy employs two complementary components: **(1) Depth Scaling.** For a hand pose in camera coordinate $m_c = \{\beta, \theta, R_c, \tau_c\}$, where $\tau_c$ denotes the wrist's 3D position and $R_c \in \mathbb{R}^{3\times3}$ is the rotation matrix, we perturb human hand's depth by randomly scaling depth $\tau_c^{z'}$ by $\lambda_s$, yielding $\tau_c^{z'} = \lambda_s \cdot \tau_c^z$. The paired image is rescaled by $1/\lambda_s$ to maintain weak-perspective consistency. $\lambda_s$ is constrained to plausible ranges to avoid unrealistic perspective distortions caused by the non-negligible physical size of the hand. **(2) In-Plane Rotation.** To improve hand position diversity in the image plane, we rotate the hand around the camera's Z-axis by a uniformly sampled angle $\varphi$, updating wrist position $\tau'_c = R_z(\varphi) \cdot \tau_c$ and global rotation $R_c' = R_z(\varphi) \cdot R_c$, where $R_z(\varphi)$ is the rotation transform matrix. The image is rotated synchronously by $\varphi$, and all transformed frames are resized to a target resolution to preserve weak-perspective projection integrity.

While our current approach focuses on perspective spatial alignment, its underlying motivation extends to a broader notion of 'physical space alignment' that aims to unify visual observations with richer physical cues. See Appendix C for additional discussion.

## 3.3 POST-TRAINING FOR DEXTEROUS MANIPULATION

Learned rich behavior priors from hand motion pretraining, Being-H requires adaptation to bridge the human-robot kinematic gap. To better address this gap, the post-training stage serves as the point where pre-trained high-level human priors are grounded into the robot's embodiment. For a general downstream embodiment support, we do not directly utilize hand motion generation. Instead, we map the hand motion latent to the robot action space, allowing the model to adapt high-level priors to low-level robot actuations. In this paper, we employ a **non-autoregressive** projection (Figure 2) for manipulation post-training and inference.

Added to the visual-text sequence, a chunk of learnable queries $(\mathbf{q}_1, \ldots, \mathbf{q}_{N_a})$ attend to the instructional context through the VLA backbone. The last hidden states of the learnable queries is viewed as the queried output, which is mapped to an executable action space with a regression MLP head $f_r$. Additionally, a lightweight MLP $f_p$ maps the robot's proprioceptive states into the VLA's embedding space and combined into the context for assistance. Formally, the conbined unified context is

Figure 4: Dataset Overview. **(left)** Scenes and tasks from different data types. **(middle)** Distribution of different data sources, data types, and durations. **(right)** Samples from different data types.

$[\texttt{visual}, \texttt{text}, f_p(\mathbf{p}_t), \mathbf{q}_{1:N_a}]$. Denote the last hidden state of action query $\mathbf{q}_i$ as $h_i$. We conduct imitation learning, minimizing the L1 loss between predicted actions $a$ and expert demonstrations $\mathbf{a}^* = \{\mathbf{a}_i^*\}$:

$$\mathbf{a}_i = f_r(h_i), \quad \mathcal{L}(\Theta_\mathrm{a}) = \frac{1}{N_a} \sum_{i=1}^{N_a} \|\mathbf{a}_i - \mathbf{a}_i^*\|_1, \tag{5}$$

where $N_a$ is the action chunk size and $\Theta_\mathrm{a}$ denotes post-training parameters, including pretrained VLA $\Theta$, action queries $\mathbf{q}_{1:N_a}$, proprioceptive projector $f_p$, and policy head $f_r$. This approach adapts the VLA to generate robot controls while preserving its cross-modal capabilities. The post-training only requires the pretrained VLA backbone and two MLP projectors, while the modules in hand motion tokenization and spatial alignment are no longer involved. We adopt this simple MLP-based projector to provide a clean evaluation of the pretrained VLA. In fact, stronger adapters could further improve downstream performance, and an alternative is discussed in Appendix B.4.

## 4 UNIHAND: SCALING UP HAND MOTION INSTRUCTIONS

To support explicit VLA pretraining from large-scale human videos, we curate UniHand, a dataset aggregated from 11 sources with detailed hand motion annotations and RGB video. It contains over 440K task trajectories (130M frames, 1,100+ hours), ensuring high diversity and coverage of real-world scenarios. Due to computational constraints, we sample 2.5M instruction data points via a balanced strategy to preserve task and source diversity, which we refer to as `UniHand-2.5M` (Figure 4). We adopt several key steps to form the curation pipeline: **(1) Hand Pose Standardization.** Our method standardizes all hand motions as MANO parameters to eliminate camera system variance and learns an explicit 2D-to-3D mapping. **(2) Task Description Labeling.** We annotate UniHand via a hierarchical labeling framework with chunk-level and per-second labels to enrich sparse texts and strengthen vision-language-motion alignment. **(3) Instructional Data Generation.** We create diverse task types to construct instruction-following data for our VLA pretraining. Each task type contains 20 base templates, expanded by Gemini-2.5-Pro to generate variants. We further use rule-based instantiation to populate these templates with grounded instructions, motion tokens, and length constraints. More curation details and dataset statistics are posted in Appendix D.1 and D.2.

## 5 EXPERIMENTS

### 5.1 EXPERIMENTAL SETUP

**Implementation Details.** We encode motion sequences with a temporal downsampling $\alpha = 4$, zero-padded to length multiple of $\alpha$ to prevent information loss. Our part-level tokenizer uses an 8-layer GRQ architecture with a group size $n = 2$, converting each one-second motion sequence into $2nL\lceil T/\alpha \rceil = 128$ tokens. Codebook sizes are $K_w = K_f = 4096$ per part shared across all RQ layers with code dimension of $d = 512$. The tokenizer is optimized with batch size 2048, learning rate $2 \times 10^{-4}$ and loss weights $\lambda_1 = 0.02, \lambda_2 = 1.0$. For multimodal modeling, we employ `InternVL3` (1B/8B/14B) backbones trained on `UniHand-2.5M`. Each instance includes a $448 \times 448$ scene image and a camera-coordinate-aligned hand motion. Hand poses are represented using MANO-D162, sampled at 15 FPS and discretized into 128 tokens per hand per second. During pertaining,

vocabulary-level logit masking with probability $\mathcal{P} = 50\%$ and token-level loss filtering within $[15\%, 95\%]$ are applied. We use AdamW with a learning rate $1 \times 10^{-5}$, batch size 128, and training on $32 \times$ A800-80G GPUs, jointly fine-tuning both ViT adapter and LLM backbone. In addition, we also include a variant noted as 'Being-H (FM)' where the MLP action head is replaced with the flow-matching head used in GR00T [7]. Please refer to the Appendix B.4 for architectural details.

**Evaluation Benchmarks.** We evaluate our pretrained VLA on two benchmarks. Below, we describe the datasets, benchmarking tasks, and evaluation metrics. All details are provided in Appendix E.

**Hand Motion Modeling**: We sample 5% data of UniHand for three sub tasks: **(1) Generation** which produces a 3D hand motion sequence from a static scene image, text instruction, and duration; **(2) Prediction** which forecasts subsequent motion given an image, a short motion context, and a follow-up text instruction; **(3) Translation** which generates a text description from an image and a motion sequence. We evaluate on two splits "**head**"(EgoDex) and "**tail**"(other datasets, which constitute a sparse long-tail like TACO, HOI4D and H2O). We assess the spatial accuracy and semantic alignment for motion generation using MPJPE, MWTE, PA-MPJPE, M2T R@3 and FID. For hand motion translation, we adopt T2M R@3 and valid rate for evaluation.

**Dexterous Manipulation**: We evaluate how the pretrained priors transfer in the downstream dexterous manipulation through both simulation benchmarks and real-world tasks. Simulation benchmarks include LIBERO and RoboCasa [65]. In the LIBERO experiment, the models are fine-tuned independently on four task suites with 50 demonstrations from two camera views for each. In the Robo-Casa experiment, the models are fine-tuned on a whole set of 24 atomic tasks, and 50 demonstrations from a single left-camera view are provided for each task. Real-world tasks are conducted using a hardware setup as shown in Figure 5. We design a suite of tasks including grasping and placing (`Pick-Place-Toy`), articulated object (`Close-Toolbox`, `Close-Lid`), deformable object (`Unfold-Clothes`), and precise motion control (`Pour-Cup`).

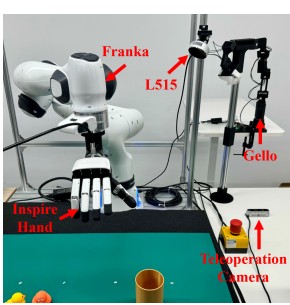

Figure 5: Hardware system.

Success is measured by a binary success rate (e.g., complete lid closure, accurate toy placement) over 20 randomized trials per task. To further assess contact-rich dexterity, we additionally introduce a `Spray-Plant` task where the robot must grasp a spray bottle by its narrow neck, stabilize it using multiple fingers, and actuate the trigger to water plants, requiring coordinated multi-finger roles and fine-grained contact control. We also report an average completion metric for the `Spray-Plant` task, computed over three ordered sub-stages. For all real-world tasks, we provide 50 demonstrations except for `Pick-Place-Toy` with 100 demonstrations to cover 4 colors of toy.

## 5.2 COMPARISONS ON HAND MOTION MODELING

To begin with, we set a one-second prediction horizon. Our evaluation first examines the model's ability to produce properly formatted motion sequences (`<MOT>......</MOT>`). As Table 1 shows, experiments on motion generation based on free-format decoding mode reveal substantial differences in valid generation rates across model scales. While Being-H-1B achieves only modest success in preserving motion block structure, Being-H-8B and -14B reach almost 100% validity, demonstrating that increased scale significantly improves structural format learning. Table 1 also evaluates

Table 1: Comparison on motion generation and translation, where we use valid rate (%) and T2M R@3 (%) as the metrics.

| Model | Valid Rate ↑ | T2M R@3 ↑ | |
| --- | --- | --- | --- |
| | | head | tail |
| ground truth | - | 33.5 | 42.7 |
| Being-H-1B | 64.8 | 12.5 | 14.3 |
| Being-H-8B | 99.8 | 18.4 | 19.7 |
| Being-H-14B | 100.0 | 19.0 | 22.1 |

motion understanding through hand motion translation. Results show larger models consistently achieve higher retrieval scores, confirming stronger bidirectional motion-language alignment.

Table 2 reports principal results on hand motion generation and prediction. We adapt GR00T N1.5 [7] as a competitive baseline following [110], redefining its action representation as dual-hand motion sequences (padding with zeros for single-hand cases). The initial hand pose serves as a state for hand motion prediction, while motion generation uses a fixed state. All models use block-formatted decoding mode for consistent evaluation, enforcing generated sequences to be decoded into the correct

Table 2: Comparison of hand motion generation and prediction tasks on both head and tail splits.

| Model | MPJPE ↓ | | MWTE ↓ | | PA-MPJPE ↓ | | M2T R@3 ↑ | | FID ↓ | |
|---|---|---|---|---|---|---|---|---|---|---|
| | head | tail | head | tail | head | tail | head | tail | head | tail |
| **# Hand Motion Generation** | | | | | | | | | | |
| GR00T N1.5 | 9.82 | 15.35 | 8.51 | 11.20 | 1.33 | 1.41 | 13.1 | 14.8 | 11.7 | 14.4 |
| Being-H-1B | 9.71 | 17.21 | 8.25 | 12.04 | 1.50 | 1.55 | 12.1 | 15.3 | 12.2 | 13.1 |
| Being-H-8B | 7.20 | 9.02 | 5.69 | 8.11 | 1.09 | 1.32 | 15.9 | 18.7 | 11.5 | 13.4 |
| Being-H-14B | 6.87 | 8.11 | 5.19 | 7.41 | 1.03 | 1.20 | 17.2 | 20.5 | 10.3 | 11.8 |
| **# Hand Motion Prediction** | | | | | | | | | | |
| GR00T N1.5 | 7.14 | 8.55 | 6.65 | 7.93 | 1.11 | 1.25 | 16.1 | 20.5 | 11.2 | 13.3 |
| Being-H-1B | 8.73 | 11.34 | 7.88 | 10.67 | 1.17 | 1.38 | 15.4 | 20.6 | 14.7 | 15.6 |
| Being-H-8B | 6.67 | 7.98 | 5.03 | 6.93 | 0.90 | 1.03 | 19.7 | 21.4 | 10.1 | 11.7 |
| Being-H-14B | 6.21 | 7.33 | 4.89 | 6.52 | 0.92 | 1.04 | 20.1 | 23.5 | 9.8 | 10.1 |

Table 3: Results of Being-H and baselines on RoboCasa manipulation tasks. We report the success rate (%) across 7 task categories and the overall average on 24 tasks.

| Task | Pick & Place | Doors | Drawers | Levers | Knobs | Insertion | Buttons | Avg. |
|---|---|---|---|---|---|---|---|---|
| GR00T N1.5 | 1.3 | 40.5 | 37.0 | 48.0 | **11.0** | 6.0 | 26.7 | 21.0 |
| InternVL3 | 1.3 | 37.0 | 36.0 | 42.0 | 9.0 | 4.0 | 18.0 | 18.2 |
| Being-H | **2.0** | **43.5** | **40.0** | **51.3** | **11.0** | **17.0** | **30.7** | **23.8** |

format. Our analysis reveals three key findings. First, larger models show superior performance with lower MPJPE, MWTE, and PA-MPJPE scores, indicating enhanced spatial grounding and more plausible pose generation. Second, they achieve better M2T R@3 and FID results, showing stronger semantic consistency between generated motions and instructions. Notably, the performance advantage is particularly pronounced on the tail split, suggesting scaling substantially improves generalization across diverse motion distributions. We further evaluate long-term motion generation capabilities in Appendix F.1.

## 5.3 Comparison on Dexterous Manipulation

**Simulation Experiments.** We evaluate Being-H in both RoboCasa and LIBERO benchmarks to assess downstream transfer in gripper-based tasks. On RoboCasa, as shown in Table 3, Being-H outperforms GR00T N1.5 in all categories and achieves a higher average success rate (23.8% vs. 21.0%) despite GR00T being pretrained on extensive robot data. The model shows superior fine-grained manipulation, especially in high-precision tasks like "Insertion" and "Buttons". Compared to InternVL3, the backbone without our pre-

Table 4: Results of Being-H and baselines on LIBERO manipulation tasks. We report success rates (%) across task categories and the overall average.

| Model | Spatial | Object | Goal | Long | Average |
|---|---|---|---|---|---|
| Diffusion Policy [21] | 78.3 | 92.5 | 68.3 | 50.5 | 72.4 |
| Octo [82] | 78.9 | 85.7 | 84.6 | 51.1 | 75.1 |
| OpenVLA [44] | 84.7 | 88.4 | 79.2 | 53.7 | 76.5 |
| $\pi$0-FAST [70] | **96.4** | 96.8 | 88.6 | 60.2 | 85.5 |
| GR00T N1.5 [7] | 92.0 | 86.0 | 92.0 | 76.0 | 86.5 |
| MolmoAct [48] | 87.0 | 95.4 | 87.6 | 77.2 | 86.6 |
| Being-H | 92.6 | 96.8 | 94.4 | 77.4 | 90.3 |
| Being-H(FM) | 95.2 | **97.0** | **97.8** | **87.8** | **94.5** |

training, Being-H achieves a +5.6% absolute gain, demonstrating a direct performance gain from our pretraining. On LIBERO, Being-H also surpasses prior VLA and imitation-learning policy baselines across all four task suites, achieving a 90.3% average success rate, while the stronger FM variant reaches 94.5% (Table 4). These results demonstrate that large-scale pretraining with human videos provides robust and transferable priors for downstream tasks. We further validate that scaling post-training demonstrations on RoboCasa leads to additional performance gains (Appendix F.4).

**Real-World Experiments.** To validate the real-world efficacy of human video pretraining, we evaluate our model on dexterous manipulation tasks (Table 5). Being-H achieves the highest success rates across all benchmarks. In contrast, the `InternVL3` baseline, which lacks physical instruction tuning and hand motion priors, exhibits significantly weaker performance. While the finetuned GR00T N1.5 performs comparably on in-domain objects for the `Pick-Place-Toy` task, its generalization degrades markedly with unseen objects and cluttered scenes. Being-H's explicit motion tokenization enables superior generalization with far less data than GR00T's implicit latent action prediction. This advantage is most evident in fine-grained manipulation. For instance, our model robustly positions

Table 5: Success rates (%) and Being-H vs. baselines on real-world dexterous manipulation tasks.

| Task | Pick-Place-Toy | | | Close-Toolbox | Close-Lid | Pour-Cup | Unfold-Clothes |
|------|------|------|------|------|------|------|------|
| | *Seen.* | *Unseen.* | *Clutter.* | | | | |
| GR00T N1.5 | **0.75** | 0.40 | 0.50 | 0.80 | 0.50 | 0.90 | 0.60 |
| InternVL3 | 0.55 | 0.55 | 0.50 | 0.50 | 0.25 | 0.55 | 0.45 |
| Being-H | **0.75** | **0.65** | **0.60** | **0.85** | **0.60** | **1.00** | **0.75** |

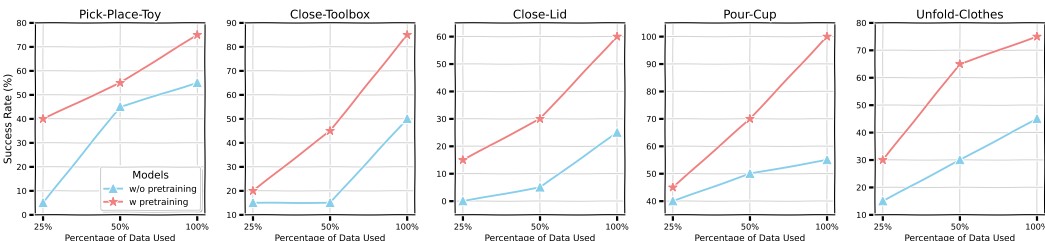

Figure 6: Comparison of data efficiency. Success rate is shown for Being-H and the InternVL3 (no VLA pre-training) when fine-tuned with varying fractions (25%-100%) of the teleoperation data.

and closes lids, pinches cloth edges to unfold fabric, and maintains a stable grasp for smooth pouring. These results underscore Being-H's successful transfer of hand motion knowledge from physical instruction tuning to downstream real-world robot control.

To evaluate data efficiency, we further compare our model against a non-pretrained InternVL3 baseline using 25%, 50%, and 100% of demonstration data across multiple tasks. As shown in Figure 6, Being-H maintains a consistent and substantial performance advantage at all data scales, demonstrating the benefit of physical instruction tuning. The hand-motion priors from pretraining enable faster adaptation, allowing our model to match or exceed baseline performance with far less data. For example, with only 25% data, our model performs comparably to the baseline trained using 100% data on `Pick-Place-Toy`, and matches the baseline at 50% on `Close-Toolbox` and `Unfold-Clothes`. In the challenging `Close-Lid` task, Being-H achieves a 15% success rate with 25% data, while the baseline fails entirely. This superior data efficiency reduces the reliance on costly teleoperated demonstrations, lowering deployment barriers for dexterous robots.

The `Spray-Plant` task presents a substantially higher level of dexterity compared to standard pick-and-place style manipulation. This requires multi-finger role allocation, stable grasp under torque, and fine-grained contact control. As shown in Table 6, Being-H achieves a markedly higher completion score (0.58) and success rate (35%) compared with the two baselines: GR00T N1.5 and InternVL3. The gap is especially pronounced in binary success, indicating that the human-pretrained priors help

Table 6: Results on the contact-rich Spray–Plant task. We report average completion score (0–1) and binary success rate (%).

| Model | Completion | Success (%) |
|-------|------------|-------------|
| GR00T N1.5 | 0.33 | 15.0 |
| InternVL3 | 0.23 | 5.0 |
| Being-H | 0.58 | 35.0 |
| Being-H (FM) | **0.63** | **40.0** |

the robot reliably achieve the coordinated grasp-and-trigger action instead of failing early due to unstable contacts. Furthermore, the flow-matching variant (Being-H (FM)) pushes performance even higher (0.63 / 40%), validating our findings in the LIBERO. These improvements highlight that the pretrained priors do not merely aid high-level planning, but also contribute to stable, multi-finger dexterous behaviors that are otherwise difficult to acquire from 50 teleoperated demonstrations alone.

### 5.4 ABLATION STUDY

**Part-Level Tokenizer Ablation.** As shown in Table 7, our results demonstrates the advantages of our part-level tokenizer over whole-hand quantization. We benchmark against two GRQ variants (4-group and 16-group RQ) with an identical codebook size ($K = K_w + K_f = 8192$), a fixed token count per second and consistent codebook dimensionality. Evaluated on UniHand's held-out test set, our method achieves superior performance, validating the benefit of separately tokenizing the hand wrist and fingers. Due to space limitation, we provide further ablation for the choices of MANO features and our hand tokenizer's impact on motion generation in Appendix F.2.

Table 7: Performance of different motion tokenization practices on the hand reconstruction task, including motion features and part-level tokenizing. Results are reported in centimeters (cm).

| Feature | Part-Level | | 4-Groups | | 16-Layers | |
|---|---|---|---|---|---|---|
| | MPJPE ↓ | PA-MPJPE ↓ | MPJPE ↓ | PA-MPJPE ↓ | MPJPE ↓ | PA-MPJPE ↓ |
| MANO-D51 | 0.556 | 0.209 | 1.165 | 0.184 | 1.466 | 0.243 |
| MANO-D99 | 0.584 | 0.149 | 1.093 | 0.148 | 1.510 | 0.170 |
| # with shape parameters $\beta$ | | | | | | |
| MANO-D109 | 0.592 | 0.160 | 1.107 | 0.140 | 1.602 | 0.201 |
| # with auxiliary joint positions $j$ | | | | | | |
| MANO-D114 | 0.523 | 0.167 | 0.810 | 0.202 | 0.996 | 0.253 |
| MANO-D162 | 0.573 | 0.129 | 0.704 | 0.138 | 1.054 | 0.226 |

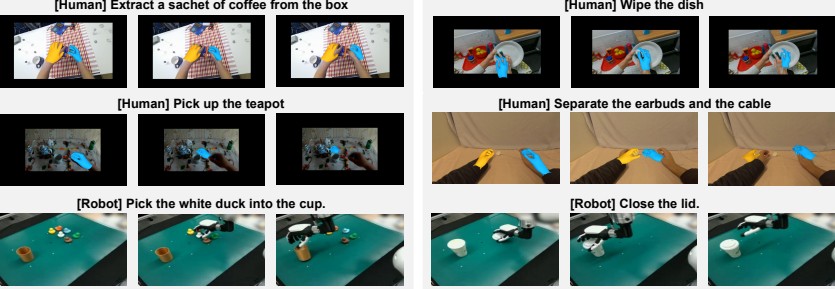

Figure 8: Qualitative examples of Being-H for generating realistic human hand motions (Row 1-2) and performing dexterous real-robot control in downstream tasks (Row 3).

**Training Data Scale Ablation.** In Figure 7, performance improves steadily with training data up to 2.5M samples, demonstrating the value of scaling diverse motion-language data. Noting that pose accuracy (PA-MPJPE) slightly drops when using 100% data, semantic alignment metrics (e.g., M2T R@3) continue to rise, We hypothesize that larger data volume increases diversity in task-object combinations and motion semantics, encouraging a shift toward prioritizing semantic plausibility over precise kinetics replication of finger pose detail. This reflects the model's growing emphasis on functional and contextual correctness as training data becomes more abundant. We discuss more ablation about data configuration in Appendix F.3.

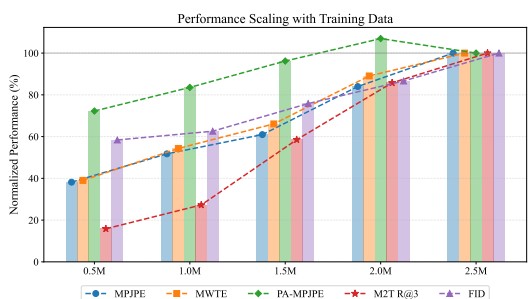

Figure 7: Being-H-8B performance scaling with training data on hand motion generation. Metrics are normalized to the final checkpoint (2.5M training samples= 100%).

## 5.5 QUALITATIVE EXAMPLES

To qualitatively demonstrate the capability of Being-H in generating physically plausible hand motions and its performance in real-robot experiments, we present representative samples in Figure 8. We provide more visualization examples in Appendix F.5

## 6 CONCLUSION

We introduce Being-H, a scalable dexterous VLA trained via physical instruction tuning. To support VLA pretraining, we curate a large-scale dataset named UniHand by integrating heterogeneous sources (motion capture, VR, RGB videos) via MANO parameter standardization. We adopt grouped residual quantization for millimeter-level accuracy and seamless integration with language models, effectively treating motion as a language. Using the human hand as a template, we transfer dexterity from human videos to robot control via mapping the VLA latent to the downstream action space, eliminating the pretraining-downstream data mismatch common in previous VLAs and achieving superior performance across downstream benchmarks.

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

# Appendix

**Roadmap.** In this section, we first introduce additional references related to this work in Section A. Then, we provide additional discussion of our pretraining and physical space alignment stage in Section B and Section C, respectively. We further provide the details of dataset statistics (Section D and evaluation setups (Section E). Finally, we carry out a thorough experiments and corresponding analysis in Section F.

TABLE OF CONTENTS

## A   ADDITIONAL RELATED WORK

This section covers related work omitted from the main text due to space constraints.

**Large Multimodal Models.** The transformer architecture [87] revolutionized language modeling [75, 76, 10], enabling powerful autoregressive text interpretation and generation. This success has extended to large multimodal models (LMMs) [112, 89, 108, 106, 105], which combine LLM reasoning [84, 85, 4] with modality-specialized encoders [77, 102] for unified multimodal understanding. Pioneering works like Flamingo [2] use cross-attention for strong few-shot VQA performance. Subsequent approaches, such as the LLaVA series [54, 53, 61], employee visual instruction tuning with curated datasets to enhance instruction-following capabilities. These datasets are often constructed by using vision models to label images, then generating QA pairs with an LLM [97, 50], or by leveraging proprietary LMMs for annotation [15, 55]. While leading LMMs [81, 23, 1] remain closed-source, recent models show growing openness through released weights [5, 80], training details [113], and data recipes [25].

**Human Motion Quantization.** Autoregressive models, which excel in long-term motion modeling and textual reasoning, commonly rely on motion quantization to represent continuous motion as discrete tokens. Widely used techniques include VQ-VAE [86, 104], Residual Quantization (RQ) [47, 36], Hierarchical Quantization (H2VQ) [98, 60], and more recent lookup-free methods [63, 91, 99]. By fine-tuning LLMs on these tokenized motions, such approaches [92, 40] achieve strong performance in intention understanding and motion generation [17, 111, 52]. This paradigm contrasts with diffusion-based models, which focus on high-fidelity motion synthesis [83, 18].

## B   ADDITIONAL DETAILS OF PRETRAINING

### B.1   MULTIMODAL INTEGRATION

Like traditional LLMs, Being-H employs next-token prediction for generation during pretraining, unifying three modalities — RGB vision, text, and hand motion — by tokenizing each into discrete tokens. While text processing follows standard LLM practices, we detail the vision and motion tokenization below:

**Vision Token.** To handle variable-resolution images and dynamic content, visual inputs undergo adaptive processing. Given an input image, we first apply dynamic patching, generating $N$ patches based on image content complexity. Following InternVL [113], a thumbnail $I_{\text{thumb}}$ (downsampled with a pixel-shuffle ratio of 0.5) is retained alongside the detailed patches to preserve global context. Features are extracted from patches and the thumbnail via a vision encoder, then projected into a unified embedding space using a MLP. Vision tokens are structured with boundary markers  and </IMG>, while <IMG_CONTEXT> acts as a placeholder dynamically replaced by actual visual embeddings during processing.

**Motion Token.** Motion data is quantized prior to integration. Given a motion feature sequence $\mathcal{M}$, the motion tokenizer discretizes it into a sequence of tokens $\{m_i\}$, structured with boundary tokens <MOT> and </MOT>. Each motion block contains of 128 tokens per second, ensuring motion information is clearly delineated within the token stream while maintaining compatibility with the transformer architecture.

**Multimodal Fusion.** All modalities are processed in a unified token space using shared embeddings and attention. During fusion, vision tokens replace <IMG_CONTEXT> placeholders, while motion tokens are inserted as structured blocks within the text sequence, forming a combined token sequence $\mathbf{S} = \{s_i\}$ where each element $s_i$ may represent text, visual, or motion content. Cross-modal attention is applied simultaneously across all modalities. For the concatenated multimodal hidden states $\mathbf{H}_{v,t,m} = [\mathbf{H}_v; \mathbf{H}_t; \mathbf{H}_m]$ (representing vision, text, and motion embeddings), we compute query, key and value through shared projections:

$$\mathbf{Q}_{v,t,m} = \mathbf{W}_Q \mathbf{H}_{v,t,m}, \quad \mathbf{K}_{v,t,m} = \mathbf{W}_K \mathbf{H}_{v,t,m}, \quad \mathbf{V}_{v,t,m} = \mathbf{W}_V \mathbf{H}_{v,t,m} \tag{6}$$

where $\mathbf{W}_{\{Q,K,V\}}$ denotes the weight matrices. This design enables direct cross-modal attention, capturing rich interdependencies between modalities, such as associating visual observations with specific hand motions, or grounding language instructions in corresponding movement sequences.

As shown in Figure 1, our pretraining extends the original vision-text parameters $\Theta_{v,t}$ to include motion parameters $\Theta_m$, facilitating unified multimodal processing via shared attention. The model thereby learns to generate coherent motion tokens conditioned on visual and linguistic context.

## B.2 HAND MOTION TOKENIZATION

As we introduced, we represent hand pose using the 3D model MANO [79], parameterized as $m = \{\theta, \mathbf{r}_{rot}, \tau, \beta\}$. An effective and efficient representation of this pose is critical for modeling motion. This paper explores five alternative feature spaces derived from the MANO parameters:

- **MANO-D51**: A 51-dimensional vector $m \in \mathbb{R}^{51}$, comprising axis-angle rotations for joints $\theta \in \mathbb{R}^{15 \times 3}$, global rotation $\mathbf{r}_{rot} \in \mathbb{R}^3$ and and translation $\tau \in \mathbb{R}^3$.

- **MANO-D99**: A 99-dimensional vector $m \in \mathbb{R}^{99}$ which replaces the axis-angle rotations in MANO-D51 with more robust 6D rotations: $\theta \in \mathbb{R}^{15 \times 6}$ and $\mathbf{r}_{rot} \in \mathbb{R}^6$.

- **MANO-D109**: This 109-dimensional representation extends MANO-D99 by incorporating the shape parameters $\beta \in \mathbb{R}^{10}$.

- **MANO-D114**: This 114-dimensional representation extends MANO-D51 by adding 3D joint positions $j \in \mathbb{R}^{21 \times 3}$. The joint positions serve only as auxiliary features during training; at evaluation and inference, only the core 51 parameters are used.

- **MANO-D162**: This 162-dimensional representation extends MANO-D99 by adding 3D joint positions $j \in \mathbb{R}^{21 \times 3}$.

Our experiments reveal that 6D rotation features yield superior reconstruction quality for finger joints, while axis-angle features are more effective for the wrist. We attribute this to the distinct structural characteristics of different hand parts. The wrist exhibits larger but simpler rotations, where the compactness and computational efficiency of axis-angle formulations are advantageous [11, 62]. In contrast, the finer, more complex motions of finger joints are better captured by the continuity and numerical stability of the 6D representation. Although the axis-angle features achieve a lower overall reconstruction error due to the dominant scale of wrist pose errors, we select the 6D rotation feature for our hand motion tokenizer based on its superior performance in Being-H training. We hypothesize that that wrist pose patterns are relatively easier for the LMM to learn, whereas accurately modeling fine-grained finger movements presents a greater challenge. Consequently, we adopt the MANO-D162 as the feature for hand motion in this work.

## B.3 TRAINING DETAILS

**Motion Tokenizer Training.** Given a hand motion sequence $m \in \mathbb{R}^{T \times D}$ represented using the MANO-D162 features, the motion tokenizer encodes each one-second window into a feature map $z \in \mathbb{R}^{T/\alpha \times d}$, followed by a multi-stage residual vector quantization (RVQ) process described in Section 3.1. For each group $g \in \{1, \ldots, n\}$, the $L$-stage RVQ produces quantized codes $\hat{z}_i^{(g)} = \sum_{\ell=1}^{L} q_\ell^{(g)}$ with residual updates $r_\ell^{(g)} = r_{\ell-1}^{(g)} - q_\ell^{(g)}$, where $r_0^{(g)} = z^{(g)}$ and $q_\ell^{(g)} = \operatorname{argmin}_{c \in C^{(g)}} \|r_{\ell-1}^{(g)} - c\|_2$. To train the tokenizer, we minimize a combination of reconstruction, commitment, and wrist-specific losses:

$$\mathcal{L} = \mathcal{L}_{\text{recon}} + \lambda_1 \mathcal{L}_{\text{commit}} + \lambda_2 \mathcal{L}_{\text{wrist}}. \tag{7}$$

- **Reconstruction Loss.** Let $\hat{m}$ denote the decoded motion sequence obtained from the quantized codes. The reconstruction loss encourages accurate recovery of the continuous MANO parameters:

$$\mathcal{L}_{\text{recon}} = \|m - \hat{m}\|_2^2. \tag{8}$$

- **Commitment Loss.** Following standard VQ-VAE training, we employ a commitment loss to stabilize the codebook usage:

$$\mathcal{L}_{\text{commit}} = \frac{1}{n \cdot L} \sum_{l,g} \Big( \|\operatorname{sg}(r_l^{(g)}) - q_l^{(g)}\|_2^2 + \beta \|r_l^{(g)} - \operatorname{sg}(q_l^{(g)})\|_2^2 \Big), \tag{9}$$

where $\operatorname{sg}(\cdot)$ denotes the stop-gradient operator.

- **Wrist Loss.** As wrist orientation and translation exhibit broader spatial variation than finger joints, we introduce an additional wrist loss that focuses explicitly on the global hand pose:

$$\mathcal{L}_{\text{wrist}} = \left\| w - \hat{w} \right\|_2^2, \tag{10}$$

where $w = [r_{\text{rot}}, \tau]$ denotes the wrist parameters and $\hat{w}$ are the corresponding decoded estimates. This term improves the numerical stability of 6D rotations and yields more reliable trajectory structure for downstream manipulation.

- **Codebook Optimization.** Each RVQ codebook $C^{(g)}$ is updated using an exponential moving average (EMA) strategy to ensure stable cluster assignments. Following prior GRQ-based tokenizers, we constrain the update magnitude to avoid codebook collapse and encourage balanced token utilization.

- **MANO Feature Preprocess.** We sample hand motion sequences at 15 FPS and tokenize them using fixed one-second windows. Since camera shift may occur within these windows and Being-H does not predict camera motion during inference, we transform each sequence into the coordinate system of its first frame. To support coherent longer-sequence generation, where each one-second segment within a multi-second output must be relative to the entire sequence's initial frame, we employ a specialized training strategy: for each one-second sample, we randomly select a reference frame from a larger 10-second window and transform the motion relative to it. This enables motion tokens to represent movements relative to varying world coordinate systems while maintaining long-term consistency.

**VLA Pretraining.** The model is trained using standard next-token prediction. To optimize the integrated motion codes, we introduce a dual-level masking strategy that operates at both the vocabulary and token levels:

- **Vocabulary-level Logit Masking.** Since motion codes $\mathcal{V}_{\text{motion}}$ constitute a small subset of the full vocabulary $\mathcal{V}$, we mask non-motion logits for motion labels with probability $\mathcal{P}$. This focuses gradient updates on the motion embedding space and prevents dilution by irrelevant tokens. For predicted logits $\mathbf{z} \in \mathbb{R}^{|\mathcal{V}|}$, we apply masking as:

$$\tilde{\mathbf{z}}_i = \begin{cases} \mathbf{z}_i & i \in \mathcal{V}_{\text{motion}} \\ -\infty & \text{otherwise.} \end{cases} \quad \text{(with probability } \mathcal{P}). \tag{11}$$

- **Token-level Loss Masking.** The token-wise cross-entropy losses are computed using the masked logits $\tilde{\mathbf{z}}$. To handle variations in motion complexity (e.g., static poses vs. unpredictable jitters), we filter extreme loss values, focusing learning on moderately challenging tokens. For per-token losses $L = \{\ell_1, \ldots, \ell_N\}$, the filtered loss set as:

$$\tilde{L} = \left\{ \ell_i \in L \mid Q_{\text{low}} \le \ell_i \le Q_{\text{high}} \right\} \tag{12}$$

where $Q_{\text{low}}, Q_{\text{high}}$ are preset percentile thresholds. The final motion loss is the mean over the filtered losses:

$$\mathcal{L}_{\text{motion}} = \frac{1}{|\tilde{L}|} \sum_{\ell_i \in \tilde{L}} \ell_i \tag{13}$$

## B.4 DOWNSTREAM ADAPTATION ALTERNATIVE

While the main paper adopts a lightweight MLP projection head $f_r$ to map the VLA's action-query embeddings to robot actions, our framework is not restricted to this simple adapter. In settings where the embodiment gap is larger or the downstream motion dynamics are more complex, a more expressive adapter may further improve performance. For this reason, we additionally explore a *flow-matching action head* as an alternative to the MLP head during post-training.

**Flow-Matching Head.** Given the final hidden states of the action queries $\{h_i\}_{i=1}^{N_a}$ from the pretrained VLA backbone, we condition a diffusion-style network $V$ on the embeddings together with the robot's proprioceptive state $p_t$. For a ground-truth action chunk $A_t$, a noised version is constructed as

$$A_t^\tau = \tau A_t + (1 - \tau)\,\epsilon, \tag{14}$$

where $\tau \in [0, 1]$ is a noise level and $\epsilon \sim \mathcal{N}(0, I)$. The flow-matching objective supervises the network to predict the denoising vector field:

$$\mathcal{L}_{\text{FM}} = \mathbb{E}_{\tau, A_t, \epsilon}\left[\left\|V(\{h_i\}, A_t^\tau, p_t) - (\epsilon - A_t)\right\|_2^2\right]. \tag{15}$$

During inference, a randomly initialized action chunk is iteratively denoised via forward Euler integration using $V_\theta$, resulting in executable robot commands. This procedure injects high-level behavioral priors encoded in the VLA backbone into the generation of low-level robot actions.

**Relationship to the MLP Adapter.** The flow-matching head preserves the same pretrained VLA backbone and the action-query interface. **The only change lies in how the final action embeddings $\{h_i\}$ are mapped to continuous robot controls, which is orthogonal to our whole pipeline.** Compared with the lightweight MLP adapter, the flow-matching head introduces several conceptual advantages. (1) It offers *greater representational expressiveness*, as the diffusion-based denoising formulation can model complex, multi-modal action distributions that a simple regression head may fail to capture. (2) The iterative refinement process offers *richer temporal and force-dependent modeling*, which is particularly beneficial for tasks that involve fine-grained interaction, continuous contact modulation, or larger embodiment discrepancies between human and robot. (3) The GR00T-style flow-matching head naturally carries *significantly more parameters* than a simple MLP projector, further strengthening its ability to fit complex action manifolds and transfer the pre-trained embeddings. These benefits come at the cost of increased computational overhead and a more involved inference procedure, but they provide a practical avenue for improving downstream performance when higher-capacity adaptation is desired.

## C  ADDITIONAL DISCUSSION OF PHYSICAL SPACE ALIGNMENT

Our pretraining bridges the vision-action gap to create a foundation VLA, but faces unique alignment challenges beyond standard visual instruction tuning. The key difficulties arise from three aspects: (1) The visual inputs from multiple sources vary in camera intrinsics and are captured under dynamic world coordinates. (2) The model's backbone is initialized with 2D vision-text pretraining, leaving it without crucial 3D spatial priors. (3) Essential physical properties, like force and friction, which humans intuitively understand, are inherently missing in video data. Unlike biological vision systems that organically develop 3D perception through embodied experience, we explicitly align these disparate data sources via perspective spatial alignment — unifying observations in a consistent coordinate system to instill 3D reasoning and physical understanding.

Beyond the two strategies proposed in Section 3.2, we believe that integrating richer physical cues can further improve the model's understanding of spatial and physical environments, which will extend 'perspective spatial alignment' to broader 'physical space alignment'. For instance, incorporating visual depth information, tactile feedback, or other multi-sensory signals may provide more grounded and embodied representations of human activities. These modalities offer complementary perspectives on physical interactions and 3D structure, which are often ambiguous or underspecified in 2D visual inputs alone.

Such multi-sensory integration could address fundamental limitations inherent in vision-only approaches. Depth information from RGB-D sensors could resolve spatial ambiguities that arise from weak-perspective projection, while tactile feedback could capture crucial contact dynamics, grip forces, and material properties that are invisible in visual observations but essential for successful manipulation. Audio signals from object interactions could further disambiguate manipulation strategies that appear visually similar but involve different physical processes, such as distinguishing between gentle placement and firm pressing actions.

These enhanced alignment strategies could create more robust representations that better capture the rich physical understanding humans naturally possess during manipulation tasks. As we scale our approach to larger and more diverse datasets, incorporating such multi-modal physical cues will become increasingly important for bridging the gap between human demonstration data and reliable robotic deployment across varied real-world scenarios.

## D  ADDITIONAL DETAILS OF UNIHAND

### D.1  DATA CURATION STEPS

**Hand Pose Standardization.** Our model learns an explicit mapping from 2D visual observations to 3D coordinates by standardizing all annotations into the MANO parameter format, ensuring both geometric precision and visual-semantic consistency. For datasets with mocap or SLAM-tracked labels, we extract MANO parameters directly [79]. When only 3D joint positions are available, we fit MANO parameters via gradient-based optimization. For datasets lacking 3D annotations, we employ HaMer [68] for frame-wise pose estimation, followed by post-processing: we detect and correct left-right hand mismatches by identifying pose discontinuities and apply temporal interpolation to fill minor gaps. This fitting process incorporates joint angle constraints and temporal smoothness to ensure physically plausible motions.

**Task Description Labeling.** We adopt hierarchical labeling to establish strong semantic grounding between vision, language, and motion, enriching the sparse or imprecise texts in existing datasets. Videos are segmented into non-overlapping chunks with a maximum length of 10 seconds, each capturing a distinct task phase. We sample frames at 2FPS and use Gemini-2.5-pro [23] for multi-scale annotation. (1) Chunk Level: We produce imperative instructions and concise summaries to describe overarching hand activities and object interactions. (2) Per-second Level: We further divide each chunk into overlapping 1-second windows, annotating them with precise captions detailing contact states, object attributes, motion trajectories relative to the camera perspective. For completeness, we separately annotate global two-handed and individual hand actions. This strategy ensures comprehensive, consistent coverage from high-level objectives to fine-grained interactions.

**Instructional Data Generation.** Building on our systematic annotations, we construct instruction-following data to establish explicit vision-language-motion alignment for our VLA. Our instruction tasks focus on multiple grounding aspects for hand motion understanding, including spatiotemporal alignment of hand trajectories with visual context, object attributes and contact states, action intentions, and consistency between high-level instructions and fine-grained motion. Guided by these principles, we develop three complementary task types. (1) Instructional motion generation: producing step-by-step motion sequences from scene images and instruction; (2) Hand motion translation: converting motions and visual cues into language descriptions; and (3) Hand motion prediction: anticipating subsequent motions given prior history, current observation, and optional instructions or task goals. For implementation, we design 20 base templates per task type, using Gemini-2.5-Pro to generate diverse variants. Each template incorporates explicit duration specifications for variable temporal granularity. We use rule-based instantiation to populate these templates with grounded instructions, motion tokens, and length constraints.

### D.2  DATA STATISTICS

As we mentioned in the main paper, we curate our dataset from three primary sources, each offering distinct advantages and trade-offs: **(1) Motion capture datasets** [28, 29, 46, 59] provide high-precision 3D annotations from multi-view systems in controlled environments (e.g., studios or labs), though they often lack diversity. For instance, OAKINK2 [103] offers multi-view, object-centric recordings of real-world bimanual manipulation involving complex tasks. **(2) VR-recorded datasets** use devices like the Apple Vision Pro with calibrated cameras and SLAM-based tracking to capture natural hand–object interactions in less constrained settings while maintaining reliable 3D ground truth. A notable example is EgoDex [38], which includes up to 194 household manipulation tasks such as tying shoelaces and folding laundry. **(3) Pseudo-annotated datasets** [69] employ off-the-shelf hand motion predictors [69] to generate

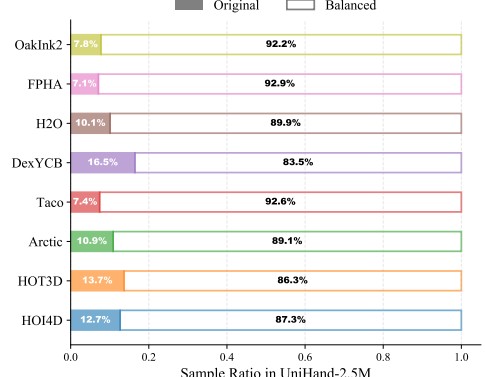

Figure 9: Comparison of instruction sample proportions in `UniHand-2.5M`: original data versus view-balanced data.

pseudo 3D labels from in-the-wild videos. Although noisier, these datasets offer superior scalability and diversity, as demonstrated in large-scale applications [12]. For example, Taste-Rob [107] contains approximately 100K egocentric videos with aligned language instructions recorded from a fixed viewpoint. The recipe of our dataset UniHand is shown in Table 8. Our dataset is aggregated from diverse sources thus provides high diversity and broad coverage of real-world scenarios. Using an instructional labeling pipeline, we generate over 165M motion–instruction pairs for dexterous VLA learning. Our pipeline supports both scalable VR-recorded data and diverse pseudo-annotated in-the-wild videos. Due to computational constraints, we sample 2.5M instruction pairs via view-invariant motion distribution balancing across tasks and sources, forming the `UniHand-2.5M` subset. The view-balanced sample proportions are illustrated in Figure 9.

Table 8: Statistics of UniHand. Our dataset is the largest egocentric hand motion dataset to date, aggregating hand motions from 11 benchmarks across three sources: motion capture, VR-recording, and pseudo-annotation. By treating human hands as the template for dexterous manipulators or grippers, we anticipate UniHand will serve as a foundational resource for VLA learning. **#Inst** refers to the number of generated instructional samples.

| Dataset | #Inst | #Seq | #Avg len | #Frames | #Hours | Hand joint | Hand Pose | Ann Granularity |
|---------|-------|------|----------|---------|--------|------------|-----------|-----------------|
| ARCTIC [28] | 17.9M | 0.3K | 725.3 | 245K | 2.3 | ✓ | ✓ | Action |
| FPHA [29] | 798K | 1.2K | 89.8 | 105K | 1.0 | ✓ | ✗ | Action |
| HoloAssist [90] | 8.0M | 2.2K | 8081.7 | 17.1M | 166 | ✓ | ✗ | Segment |
| H2O [46] | 3.7M | 0.9K | 121.5 | 115K | 1.1 | ✓ | ✓ | Action |
| HOI4D [59] | 21.2M | 3.0K | 273.0 | 825K | 7.6 | ✓ | ✓ | Action |
| HOT3D [6] | 8.7M | 2.8K | 150.0 | 420K | 3.9 | ✓ | ✓ | N/A |
| OAKINK2 [103] | 18.5M | 2.8K | 244.4 | 695K | 6.5 | ✓ | ✓ | Action |
| TACO [58] | 11.5M | 2.2K | 154.0 | 340K | 3.2 | ✓ | ✓ | Action |
| DexYCB [14] | 3.6M | 5.6K | 72.8 | 410K | 3.8 | ✓ | ✓ | N/A |
| Taste-Rob [107] | 1.9M | 85K | 164.1 | 14M | 130 | ✗ | ✗ | Trajectory |
| EgoDex [38] | 70.6M | 338K | 264.8 | 89.6M | 829.4 | ✓ | ✗ | Trajectory |
| Total | 166.5M | 444.1K | - | 130M | 1155 | ✓ | ✓ | Fine-Grained |

# E  ADDITIONAL EVALUATION SETUPS

## E.1  HAND MOTION MODELING

**Dataset Setups.** We reserve 5% of videos along with paired annotations from UniHand for evaluation. Wrist translation distributions vary across sources, with EgoDex as the dominant contributor. We augment other sources to balance translation space coverage, resulting in a distribution where EgoDex forms the central mode and other datasets constitute a sparse long-tail. To reflect this structure, we evaluate on two distinct splits: the "**head split**"(held-out EgoDex samples) and the "**tail split**"(TACO, HOI4D, H2O, and OakInk2), assessing both dominant pattern capture and generalization to less frequent motion contexts.

**Evaluation Metrics.** We employ multiple metrics to holistically evaluate the model's ability to generate physically plausible, temporally coherent, and instruction-faithful hand motions, which are detailed below:

- **MPJPE (Mean Per Joint Position Error).** measures overall spatial accuracy by computing the mean Euclidean distance between each generated joint and its ground-truth in absolute 3D space.

- **MWTE (Mean Wrist Translation Error).** evaluates global trajectory fidelity through the mean Euclidean distance between predicted and ground-truth wrist positions across the sequence.

- **PA-MPJPE (Procrustes Aligned MPJPE).** isolates relative pose accuracy by aligning predicted joints to the ground truth via rigid transformation (including scaling, rotation, and translation).

- **M2T R@3 (Motion-to-Text Retrieval Top-3 Accuracy).** assesses semantic alignment by embedding generated motion into a shared representation space and retrieving the top-3 matching descriptions using a dataset-specific text-motion retrieval model (TMR [71]).

- **FID (Fréchet Inception Distance).** quantifies the distribution similarity by comparing the generated and real motion embeddings in the dataset-specific latent space of the TMR model, measuring how well synthesized motions match the true data distribution.
- **T2M R@3 (Text-to-Motion Retrieval Top-3 Accuracy).** reports how well generated descriptions retrieve corresponding motions from a database, which verifies whether the model's text output accurately captures the semantic content of motion.
- **Valid Rate.** We supplement T2M R@3 with the valid generation rate for free-form generation, which quantifies how consistently the model produces motion sequences adhering to the required structural format.

### E.2 REAL-WORLD DEXTEROUS MANIPULATION

**Robot System.** We use a Franka Research 3 arm (7-DoF), Inspire hand (6-DoF), and RealSense L515 camera for RGB streaming. To collect demonstrations for imitation learning, we introduce an improved teleoperation system that integrates a Gello exoskeleton [93] for arm control with a RealSense D435i camera for hand pose estimation and retargeting [73, 101] (Figure 5).

**Evaluation Tasks.** We design a suite of real-world manipulation tasks — including grasp-and-place, articulated object interaction, and deformable object manipulation — to evaluate fundamental skills, generalization, and precision. For each task, we collect 50–100 teleoperated trajectories to post-train our VLA. The policy maps egocentric RGB images and proprioception to action chunks containing end-effector poses and hand joint positions.

- **Grasping and Placing (`Pick-Place-Toy`):** This task comprises three scenarios of increasing difficulty. The **Seen** scenario evaluates foundational grasping of a familiar toy. The **Unseen** scenario tests object generalization by introducing a novel toy with distinct properties (e.g., color). Finally, the **Clutter** scenario assesses advanced perception and planning, requiring the robot to locate and retrieve the target from a cluttered array of distractors.
- **Articulated Object Manipulation (`Close-Toolbox`, `Close-Lid`):** These tasks require precise closure actions on a toolbox and a cup lid. They rigorously evaluate the model's capability for accurate end-effector positioning, orientation, and stable interaction with articulated object mechanics.
- **Deformable Object Manipulation (`Unfold-Clothes`):** This task challenges the robot to unfold a piece of cloth, testing its ability to perform fine-grained, multi-finger manipulation and reason about the dynamic state of non-rigid objects.
- **Precise Motion Control (`Pour-Cup`):** This task assesses motion planning and dynamic control, requiring the generation of continuous, stable trajectories to pour liquid from one cup to another while maintaining temporal action coherence.
- **Contact-Rich Dexterous Manipulation (`Spray-Plant`):** This task evaluates the policy's ability to execute coordinated multi-finger dexterity under tight embodiment constraints. The robot must grasp a spray bottle by its narrow neck using a stable multi-finger configuration, reposition it towards the plant, and actuate the trigger to spray water. Unlike simple grasp-and-place tasks, this setting requires precise finger-role allocation (e.g., thumb and ring/pinky for stabilization, index finger for actuation), continuous force modulation, and accurate 3D spatial control. The task poses significant challenges involving contact-rich manipulation, fine-grained hand articulation, and dynamic interaction with deformable mechanisms (i.e., the trigger).

**Evaluation Protocol.** We use success rate as our primary evaluation metric and benchmark our Being-H against two baselines: GR00T N1.5 [7] and InternVL3 [113]. GR00T is selected as it is a large-scale VLA uniquely pre-trained on egocentric human videos for dexterous manipulation, contrasting with gripper-centric models like OpenVLA [44]. InternVL3 provides a direct architectural and scale-matched comparison but lacks our method's hand motion pre-training and physical alignment. All models undergo identical post-training on the same teleoperation datasets to assess the benefits of our pre-training on human hand data. To better understand model capabilities and failure modes, we perform a qualitative analysis of model behavior across three dimensions: motion precision (e.g., `Close-Lid`), semantic understanding of instructions (e.g., `Pick the white duck`), and robustness in complex tasks (e.g., `Unfold-Clothes`). To further assess the model's capability in

contact-rich dexterous manipulation, we additionally report the completion score on `Spray-Plant` task. This setting stresses fine-grained control and embodiment adaptation, as successful execution requires coordinated multi-finger grasping, stable object reorientation, and precise trigger actuation. Thus, we report an average completion score by decomposing the task into three sequential sub-stages (grasping, positioning, spraying), allowing a more granular analysis of the manipulation progress.

Table 9: Comparison results of motion generation and prediction tasks upon long-range sequences. We adopt the soft-formatted decoding mode and report short-term (2–5s) and long-term (6–10s) results, respectively.

| Model | Short-Term (2–5s) | | | | | | Long-Term (6–10s) | | | | | |
|---|---|---|---|---|---|---|---|---|---|---|---|---|
| | MPJPE ↓ | | MWTE ↓ | | PA-MPJPE ↓ | | MPJPE ↓ | | MWTE ↓ | | PA-MPJPE ↓ | |
| | head | tail | head | tail | head | tail | head | tail | head | tail | head | tail |
| **# Hand Motion Generation** | | | | | | | | | | | | |
| Being-H-1B | 8.97 | 9.96 | 7.01 | 8.75 | 1.43 | 1.67 | 9.12 | 11.24 | 7.13 | 9.91 | 1.60 | 1.81 |
| Being-H-8B | 7.55 | 8.45 | 5.78 | 7.51 | 1.10 | 1.30 | 8.21 | 9.98 | 6.12 | 8.34 | 1.22 | 1.36 |
| Being-H-14B | 7.43 | 8.39 | 5.65 | 7.39 | 1.11 | 1.28 | 7.98 | 9.72 | 5.88 | 8.01 | 1.18 | 1.32 |
| **# Hand Motion Prediction** | | | | | | | | | | | | |
| Being-H-1B | 8.44 | 9.52 | 6.71 | 7.99 | 1.20 | 1.45 | 9.01 | 10.98 | 6.98 | 8.75 | 1.35 | 1.50 |
| Being-H-8B | 7.67 | 8.20 | 5.81 | 7.13 | 1.01 | 1.22 | 8.23 | 9.67 | 6.23 | 7.83 | 1.14 | 1.27 |
| Being-H-14B | 7.39 | 8.51 | 5.77 | 7.21 | 1.05 | 1.25 | 8.01 | 9.45 | 6.02 | 7.67 | 1.18 | 1.30 |

# F  ADDITIONAL EXPERIMENTS

## F.1  COMPARISONS ON LONG-RANGE MOTION GENERATION

We evaluate the long-term motion generation capabilities of different Being-H's variants in Table 9. To mitigate the inherent error accumulation that causes trajectory drift and degraded pose quality in longer sequences, we employ the soft-formatted decoding mode to constrain outputs within plausible ranges relative to ground-truth distributions. Results are categorized into **short-term** (2–5 seconds) and **long-term** (6–10 seconds) ranges to precisely examine the quality degradation. Using MPJPE, MWTE, and PA-MPJPE as the metrics for spatial accuracy and trajectory stability, we observe that generation quality deteriorates with longer sequences, evidenced by elevated MPJPE and MWTE. However, larger models maintain more stable spatial accuracy, as they better leverage the partial ground-truth context to anchor the trajectory under these soft constraints.

## F.2  ABLATION STUDY OF HAND MOTION TOKENIZATION

In this section, we provide additional exploration about the configuration of hand motion tokenization.

**MANO Feature Choice.** We analyze the impact of different motion features through GRQ reconstruction. Axis-angle features yields superior overall reconstruction accuracy (e.g., MANO-D99 vs. MANO-D51; MANO-D162 vs. MANO-D114), while 6D rotation achieves better PA-MPJPE scores, indicating its potential advantage at modeling the finger actions. As Table 7 shows, the 6D rotation-based MANO-D162 feature is most effective for Being-H. We also find that incorporating auxiliary joint positions ($j$) enhances performance, but modeling hand shape parameters ($\beta$) degrades it. Consequently, we hold shape parameters constant from each sequence's initial frame, focusing the tokenizer exclusively on motion dynamics.

**Impact of Motion Tokenizer on Hand Motion Generation.** We adopt the MANO-D162 + part-level tokenizer as our default configuration for training Being-H. To verify this choice, we benchmark against three high-performing alternatives (MANO-D114 + 4-groups, MANO-D162 + 4-groups, and MANO-D114 + part-level). As demonstrated in Table 10, our default tokenizer consistently outperforms others in generation tasks, despite a slight higher reconstruction error with larger MPJPE shown in Table 7. We attribute its effectiveness to the 6D rotation representation and part-level decomposition, which better facilitate temporal modeling and autoregressive generation of fine-grained motions.

Table 10: Ablation of motion tokenizer variants and data recipes, where "Trans" and "Pred" denote hand motion translation and prediction task respectively, while "Balance" represents view-invariant motion distribution balancing, a strategy adopted for physical space alignment. Evaluations are carried out on the hand motion generation benchmark.

| Variants | MPJPE ↓ | | MWTE ↓ | | PA-MPJPE ↓ | | M2T R@3 ↑ | | FID ↓ | |
|---|---|---|---|---|---|---|---|---|---|---|
| | head | tail | head | tail | head | tail | head | tail | head | tail |
| **# Base** | | | | | | | | | | |
| Being-H-8B | **7.20** | **9.02** | 5.69 | **8.11** | **1.09** | **1.32** | **15.9** | **18.7** | 11.5 | **13.4** |
| **# Tokenizer Variants** | | | | | | | | | | |
| MANO-D114 + 4-Groups | 8.31 | 10.35 | 6.52 | 9.11 | 1.14 | 1.35 | 13.1 | 14.7 | 13.7 | 15.6 |
| MANO-D162 + 4-Groups | 7.98 | 9.71 | 5.58 | 8.98 | 1.09 | 1.38 | 15.4 | 17.1 | 11.9 | 14.3 |
| MANO-D114 + Part-Level | 7.74 | 9.92 | 6.11 | 8.83 | 1.16 | 1.41 | 12.3 | 16.1 | 13.1 | 12.3 |
| **# Data Recipe** | | | | | | | | | | |
| w/o Trans | 7.22 | 9.11 | **5.51** | 8.12 | 1.27 | 1.46 | 13.2 | 11.6 | 13.1 | 15.7 |
| w/o Pred | 8.01 | 10.97 | 6.34 | 9.03 | 1.11 | 1.52 | 13.7 | 15.2 | **11.1** | 14.9 |
| w/o Balance | 8.54 | 12.13 | 7.74 | 10.04 | 1.24 | 1.57 | 11.3 | 10.3 | 15.8 | 16.3 |

### F.3 ABLATION OF DATA CONFIGURATION

To optimize Being-H, we construct UniHand with diverse instruction types. We also adopt the view-invariant motion distribution balancing (Section 3.2) to refine our dataset. We systematically ablate these components to understand their impact.

**Impact of view-invariant motion distribution balancing.** As described in Section 3.2, our balancing strategy equalizes motion-view coverage by augmenting hand poses under consistent weak-perspective projection. We evaluate its effect from two perspectives: **(1) Tokenizer Learning.** As shown in Figure 10, balancing significantly reduces GRQ reconstruction errors on the held-out test set of UniHand, even for datasets without explicit augmentation (e.g., EgoDex, Taste-Rob). This indicates that more precise encoding of global wrist and fine-grained finger motion due to evenly distributed motion-view coverage. **(2) Pretrained VLA Learning.** We compare Being-H with a variant trained on the dataset without the balancing strategy. Table 10 shows that removing balancing causes substantial performance degradation on the tail split. Without it, the model overfits to dominant camera configurations and fail to gen-

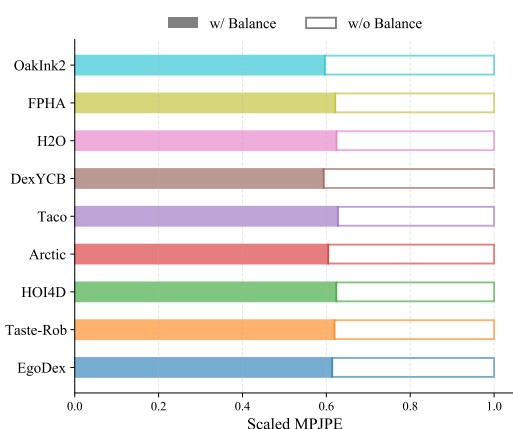

Figure 10: Ablation of view-invariant motion distribution balancing ("Balance") on the motion reconstruction task.

eralize to underrepresented perspectives. These results underscore the critical role of view-invariant balancing in enhancing both the tokenizer's representational robustness and the pretrained VLA's generalization for generating accurate and semantically grounded motions across diverse views.

**Benefits of auxiliary supervision tasks.** As introduced in Section D.1, we incorporates two auxiliary supervision types in UniHand in addition to instructional motion generation data: hand motion translation and contextual motion prediction. Evaluation on the core motion generation task reveal their distinct benefits as shown in Table 10. First, removing translation data yields marginal changes in global wrist metrics (MPJPE, MWTE), but leads to clear degradation in PA-MPJPE, M2T R@3, and FID, underscoring its role in generating semantically aligned, detailed articulations. Second, removing motion prediction data leads to uniform drops across all metrics, highlighting its central role in ensuring temporally coherent and global context awareness. Thus, auxiliary supervision not only enables task-specific capabilities but also significantly strengthens the core generation model through improved semantic and temporal grounding.

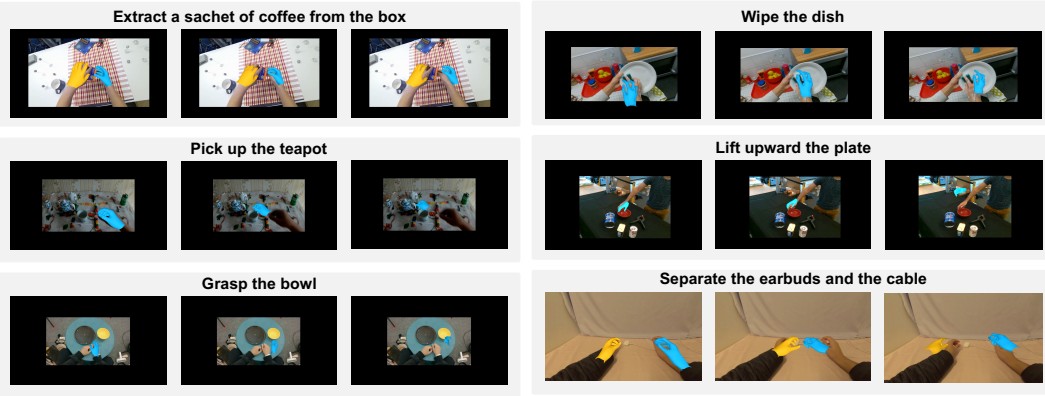

Figure 11: Qualitative examples of hand motions generated by Being-H-8B across diverse tasks, scenes, and viewpoints. Each block shows simplified instruction and three temporal frames of the motion, rendered in the first-frame camera coordinates and overlaid on the input image. Black padding ensures consistent weak-perspective projection.

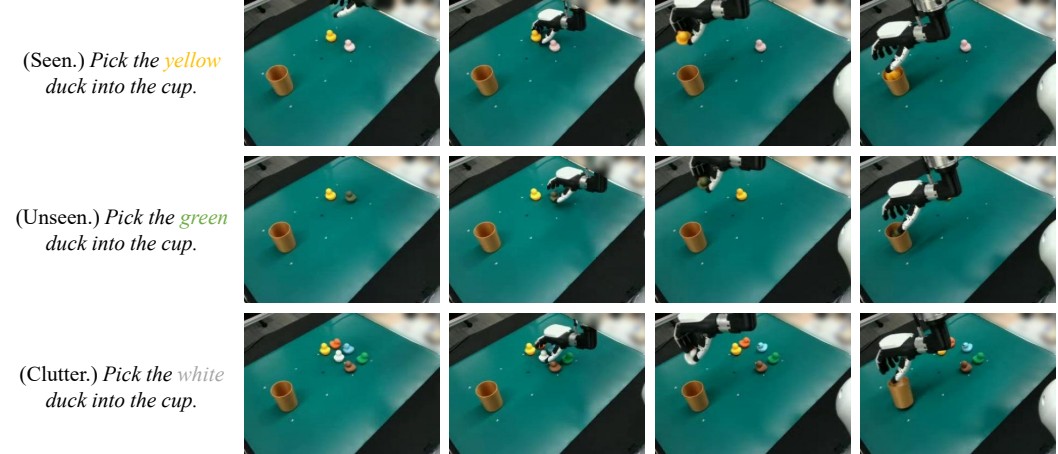

Figure 12: Qualitative examples of real-world task generation: Being-H performing the `Pick-Place-Toy` task under three conditions, including seen objects, unseen objects, and cluttered scenes.

## F.4 ADDITIONAL POST-TRAINING SCALING EXPERIMENTS

In the main paper, we demonstrate that Being-H achieves strong downstream performance under *few-shot* post-training. To investigate whether our pretrained VLA continues to improve beyond the few-shot regime, we conduct a scaling study on RoboCasa using the information-rich three-view setup and the higher-capacity flow-matching action head (Being-H(FM)). This setting provides the most expressive inputs and output head, enabling us to examine whether additional robot demonstrations can further ground the pretrained human priors and push the policy performance to a higher level.

We fine-tune Being-H(FM) with three demonstration budgets per task: 50, 100, and 200 demonstrations across the 24 RoboCasa atomic tasks, corresponding to a total of 1200, 2400, and 4800 demonstrations, respectively. Policies receive three-view RGB observations and proprioception as input. We report success rates (%) across seven task categories and the overall average.

As is shown in Table 11, scaling the amount of post-training demonstrations yields consistent and meaningful improvements. The overall average success increases from 33.5% to 38.5% and further to 42.0% as the demonstration budget grows from 50 to 200, indicating that the pretrained human priors can be progressively grounded into the robot embodiment with additional supervision.

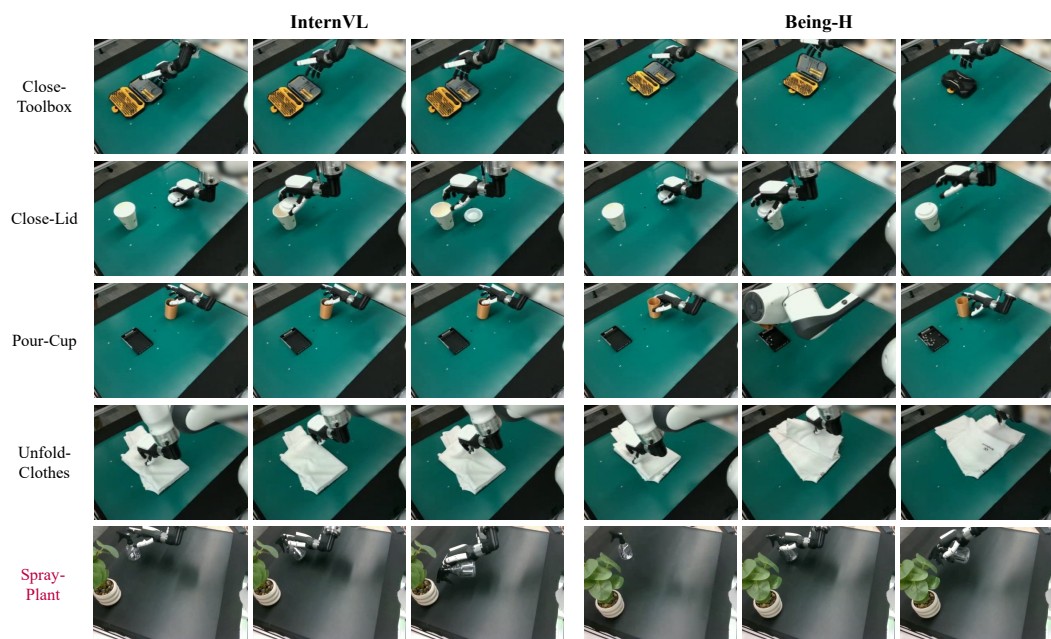

Figure 13: Qualitative comparisons between Being-H and InternVL3 baseline without VLA pretraining.

Table 11: Scaling post-training demonstrations on RoboCasa (3-view, Being-H(FM)). We report success rates (%).

| Tasks | Pick & Place | Doors | Drawers | Levers | Knobs | Insertion | Buttons | Avg. |
|---|---|---|---|---|---|---|---|---|
| 50×24 demos | 10.0 | 64.5 | 48.0 | 58.0 | 21.0 | 26.0 | 34.0 | 33.5 |
| 100×24 demos | 16.0 | 71.5 | 55.0 | 60.7 | 25.0 | 27.0 | 38.0 | 38.5 |
| 200×24 demos | 18.0 | 76.0 | 63.0 | 66.7 | 24.0 | 31.0 | 41.3 | 42.0 |

## F.5 ADDITIONAL QUALITATIVE EXAMPLES

**Hand Motion Generation.** To demonstrate Being-H's capability in generating physically plausible hand motions across diverse tasks, scenes, and viewpoints, we present additional samples in Figure 11 for visualization. Each example illustrates a generated motion sequence rendered over the first frame in its respective camera coordinate system, with hands color-coded for clarity (yellow for the left hand, blue for the right hand). To enable consistent visualization under a unified weak-perspective projection, we apply a standardized transformation to these frames, which introduces black borders around the images. The effective region — excluding the black padding — captures variations in hand-object interaction depth, where closer interactions manifest as larger apparent sizes.

**Real-world Task Generalization.** Our model adeptly handles both single-hand and dual-hand manipulations across a broad spectrum of tasks, showcasing robust generalization to varied viewpoints and physical contexts. For instance, Figure 12 highlights how Being-H not only successfully picks up the seen yellow duck but also generalizes to the unseen green duck. Even more impressively, in a cluttered environment with multiple distractors, Being-H accurately adheres to the instruction "Pick the white duck", precisely identifying and retrieving the target white duck. This underscores the model's seamless integration of visual perception, language comprehension, and action generation.

**Being-H vs. InternVL3 on Fine-grained Tasks.** The advantages of Being-H become especially evident in tasks requiring fine-grained manipulation. In contrast, the baseline model InternVL3, which lacks physical instruction tuning and prior knowledge related to hand motion dynamics, exhibits markedly weaker performance. A qualitative comparison presented in Figure 13 clearly reveals several characteristic failure modes of InternVL3:

Spatial Perception Deviations

Fine-grained Control

Figure 14: Failure case modes across real-world dexterous manipulation tasks.

- `Close-Toolbox`: The motion trajectory from InternVL3 baseline lacks precision, frequently missing contact with the edge of the toolbox lid, thereby failing to generate sufficient force to close it.

- `Close-Lid`: The InternVL3 demonstrates positional deviation, often misaligning the lid beside the cup's rim rather than seating it correctly.

- `Pour-Cup`: The grasp of InternVL3 baseline is unstable, occasionally failing to securely hold the cup, which compromises the stability of the subsequent pouring motion.

- `Unfold-Clothes`: The InternVL3 baseline misjudges the operational height, causing the fingers to close at an incorrect elevation and miss the cloth's edge, resulting in a failed unfolding attempt.

- `Spray-Plant`: The InternVL3 baseline struggles to establish a stable grasp on the bottle's narrow neck, often making contact too low or at an incorrect angle, which prevents the model from allocating proper finger roles for support. As a result, the bottle frequently slips or rotates during the trigger-pressing phase, leading to incomplete or failed watering attempts.

### F.6    FAILURE CASES STUDY

Across the evaluated tasks, we observe that the failure cases (Figure 14) of Being-H fall into two primary categories.

**Spatial perception deviations.** A number of errors originate from small but impactful inaccuracies in estimating 3D contact locations from monocular RGB observations. In tasks such as `Unfold-Clothes` and `Close-Toolbox`, the end-effector visually appears to be correctly aligned, yet the actual contact point remains slightly offset, preventing effective interaction with the cloth edge or toolbox lid. Similar but more subtle deviations occur in `Close-Lid` and `Pick-Place-Toy`, where the gripper occasionally misses the cup rim or toy by a narrow margin. These errors reflect the inherent ambiguity in depth and fine-scale geometry under single-view RGB input, which can cause near-contact states to result in failure despite seemingly correct approach trajectories.

**Fine-grained control.** Another typical kind of failure arises when the global approach and grasp region are correct, but the task requires precise finger placement or fine-grained contact modulation. This pattern appears in the duck-toy grasping and particularly in the `Spray-Plant` task. Specifically, the robot stabilizes the spray bottle in an overall correct pose, which is substantially

better than the InternVL3 baseline. However, slight deviations in finger position or closure timing occasionally prevent the index finger from actuating the trigger. While the pretrained model provides strong *high-level* behavioral priors, such priors do not fully guarantee *fine-grained* dexterous control, especially when subtle contact geometry and multi-finger coordination are required. The lightweight MLP adapter further compounds this issue, as its single-step mapping offers limited expressiveness for modeling delicate force adjustments.

Although these errors occur infrequently, they suggest that further gains may be achieved by enriching 3D perception (e.g., multi-view or depth cues) and exploring more expressive continuous-action heads to better support fine-grained dexterous control.

## G USE OF LARGE LANGUAGE MODELS

In this work, the large language model (LLM) is employed exclusively for text polishing purposes. Its role is limited to refining the linguistic quality, coherence, and stylistic consistency of the textual content, without involvement in data generation, analytical reasoning, or substantive content creation.

