# OpenReview forum: "Vision-Language-Action Pretraining from Large-Scale Human Videos"
_ICLR.cc/2026/Conference — Submitted to ICLR 2026_

### Official Review · Reviewer_QutY · 2025-10-28

**Soundness:** 2
**Presentation:** 1
**Contribution:** 2
**Rating:** 2
**Confidence:** 3

**Summary:**

This paper proposes Being-H, a dexterous Vision-Language-Action (VLA) model trained through Physical Instruction Tuning, which integrates large-scale human video data with explicit 3D hand motion modeling. The approach leverages a newly curated dataset (UniHand) and introduces a part-level motion tokenizer for millimeter-level precision.

**Strengths:**

- [S1] Physical Instruction Tuning effectively extends visual instruction tuning to the physical domain, an interesting idea that unifies VLA, physical space alignment, and post-training adaptation for robotic tasks.
- [S2] The authors conduct comprehensive analysis, including quantitative and qualitative experiments (simulation and real-robot tasks).

**Weaknesses:**

- [W1] The manuscript, in its current form, lacks sufficient clarity and is not ready for publication. The inconsistent notation and disorganized presentation of formulas significantly hinder understanding. Substantial revision and careful polishing are required to improve the logical flow and overall readability. I also have several specific questions regarding unclear points and expect the authors to address them thoroughly.

- [W2] Scalability limitations of Physical Instruction Tuning. Although the authors frame their method as scalable, it still depends on datasets with explicit 3D hand motion annotations, which are costly and non-trivial to obtain. This reliance contradicts the claimed scalability and restricts the applicability of the approach to specialized domains where such annotations exist.

- [W3] Complex pipeline with modest gain. The training and alignment processes (including GRQ-based motion tokenization and physical space alignment) add substantial system complexity, yet the improvements over strong baselines are relatively modest compared to existing VLA model, such as GR00T.

**Questions:**

- [Q1] What are $\mathcal{X}_Q$ and $y_i$? How are they related to motions $\mathbf{m}$?
- [Q2] Do $\mathbf{m}$ and $m$ indicate the same thing?
- [Q3] What is the exact form of $\mathcal{L}_{\text{recon}}$ in Equation 3?
- [Q4]  What is the exact form of $\mathcal{L}_{\text{commit}}$ in Equation 3? The manuscript does not describe the details about this objective.

---

> ### Author Response · Authors · 2025-11-22
> **Response to Reviewer QutY (1/2)**
>
> We thank the reviewer for the detailed assessment. We appreciate the concerns regarding clarity, scalability, and system complexity, and we have revised the manuscript to address these points. Below we respond to each issue.
>
> ### Clarity and notation explanation
> We thank the reviewer for their valuable feedback on improving the clarity and notation of our manuscript.
> Some explanations were shortened due to space constraints, and we have now revised Section 3 to enhance the logical flow and provide more explicit definitions.
> The key improvements are:
> 1. Clearer definition of training data format. We organize pretraining data as instructional pairs, where $\mathcal{X}_Q$ denotes the token sequence for the “question”, and $\mathcal{X}_A = \{ y_i \}$ denotes the token sequence for the “answer”. $y_i$ refers to the individual tokens in the answer sequence. This is now clearly stated in Sec. 3.
> 2. Unified notation for motion variables. In the initial submission, $\mathbf{m}$ denotes the motion modality and $m$ denotes a MANO-structured vector, which could cause confusion. We now unify them under a single notation $m$.
> 3. Explicit forms of the losses. We now explicitly claim they are reconstruction loss ($L_{recon}$) and commitment loss ($L_{commit}$) of methods in VQ-VAE style. And we include the full equations in the Appendix B.3.
>
> We believe these revisions significantly improve readability and reduce cognitive load. We are happy to provide any further clarification if needed.
>
> ### Scalability of Human Data and Clarification on 3D Annotations
> We thank the reviewer for the opportunity to clarify a key point regarding our use of human data.
> 1. **Our pretraining does not require 3D hand annotations to be physically collected during data capture.**
> A central premise of our work is that large-scale human data for pretraining is feasible and cost-effective.
> In fact, this field has emerged as a prominent area of study in the last several years.
> While our dataset includes some MoCap data with ground-truth MANO parameters, the vast majority (VR-recorded data like EgoDex and pseudo-annotated data like Taste-Rob) do not include hand motion during collection. **Instead, they are labeled using vision-based estimation methods.**
> The acquisition of this data type only requires standard RGB or RGBD cameras, and can be further applied to internet videos using open-source models (as we did on Taste-Rob).
> Therefore, scaling this data source requires only low-cost cameras or existing video repositories, not specialized motion capture systems, especially when comparing to the cost of real-robot demonstrations.
> 2. Human data provide unique manipulation priors unavailable in robot datasets. Human demonstrations capture dexterous and flexible strategies refined by real-world experience. In contrast, simulation data is limited by the sim-to-real gap, while teleoperation data is often constrained by non-ideal control interfaces, which can distort natural human motion and intent.
>
> 3. Human and robot data are complementary. We envision a future framework that co-pretrains on both human videos and multi-embodiment robot data, for instance, through multiple action heads or a unified action space. The contribution of this work is to first establish that human videos alone provide a powerful and scalable source of manipulation priors, a claim our experiments robustly validate.

---

> ### Author Response · Authors · 2025-11-22
> **Response to Reviewer QutY (2/2)**
>
> ### Complexity vs. performance gain
> We clarify several important distinctions and correct misconceptions.
>
> 1. We adopt a simpler and efficient post-training pipeline compared with GR00T. During post-training, we use only the pretrained transformer backbone and a lightweight MLP projector; **the motion tokenizer and spatial alignment are not used.** Now we explicitly strengthen this in Sec. 3.3 (see L290-292). This design choice proves that our human-video pretraining produces highly transferable priors, requiring only a simple mapping to the robot's action space.
> In contrast, GR00T's adaptation employs a more complex and costly flow-matching head. Our MLP uses the simplest possible adapter precisely to prove the effectiveness of pretraining on human videos.
>
> 2. **Extended experiments show that a stronger adapter (flow-matching) yields even better results.** To more thoroughly validate the effectiveness of Being-H, we add experiments on the LIBERO benchmark (see L341-347, L408-425) and a dexterous spray-plant real-robot task (grasping and squeezing a spray bottle to water plants; see L352-356, L462-476, L1387-1394, L1403-1408). In these experiments, we additionally report a Being-H variant with a flow-matching head (see Appendix B.4 for implementation details and a comprehensive discussion between flow-matching head and MLP projector). The results confirms that while our simple MLP successfully transfers human priors, a more powerfuly adapter yields further gains.
> This shows that the expressive capacity of the adaptation module matters. **Thus, GR00T naturally possesses an advantage due to the flow-matching head, which is orthogonal to the whole pipeline.**
>
> 4. **A Comparative View of Pretraining Complexity**
>
>     Both Being-H and GR00T handle multi-source data, but with different strategies. GR00T requires separate, embodiment-specific action heads. Our approach uses a unified spatial alignment module for perspective normalization and an independently trained motion tokenizer that converts hand motions into motion tokens.
>     After labeling the motion tokens, both the training inputs and loss calculations rely entirely on these motions throughout the process.
>     The difference --- discrete vs. continuous action representations --- represents a design choice, not a clear complexity advantage for either system.
>
>
>
> 5. **Isolating the Source of Performance Gains**.
>
>     GR00T benefits from large-scale robot pretraining, which provides robot-specific kinematics, contact mechanics, and viewpoints similar to the downstream environments.
>     However, Being-H uses no robot data in pretraining and merely a few demonstrations for adaptation, yet still surpasses GR00T.
>     To isolate the gain from human pretraining, we fine-tuned our VLM backbone InternVL3 under the same conditions.
>     Being-H's substantial improvement over InternVL3 indicates the value of our hand-motion pretraining.
>     This is most evident in our new dexterous spray-bottle task, Being-H achieves a very large margin, highlighting the irreplaceable value of human dexterity priors.
>     As emphasized earlier, robot data and human data are complementary, and our work focuses on demonstrating that human videos alone provide efficient priors.
>     We believe that introducing robot data to our VLA pretraining pipeline can further enhance performance, which is left for promising future work.
>
> The additional results mentioned above:
>
> **LIBERO Manipulation (Success Rate %)**
>
> | Model             | Spatial | Object | Goal | Long | Avg.  |
> |-------------------|---------|--------|------|------|-------|
> | Diffusion Policy  | 78.3    | 92.5   | 68.3 | 50.5 | 72.4  |
> | Octo              | 78.9    | 85.7   | 84.6 | 51.1 | 75.1  |
> | OpenVLA           | 84.7    | 88.4   | 79.2 | 53.7 | 76.5  |
> | π₀-FAST           | **96.4**| **96.8**   | 88.6 | 60.2 | 85.5  |
> | GR00T N1.5        | 92.0    | 86.0   | 92.0 | 76.0 | 86.5  |
> | MolmoAct          | 87.0    | 95.4   | 87.6 | 77.2 | 86.6  |
> | **Being-H**      | 92.6     | **96.8** | **94.4** | **77.4** | **90.3** |
> | **Being-H(FM)**  | **95.2**   | **97.0** | **97.8** | **87.8** | **94.5** |
>
> **Spray-Plant Dexterous Task**
>
> | Model        | Completion Score ↑ | Success Rate ↑ |
> |--------------|---------------------|----------------|
> | GR00T N1.5   | 0.33                | 0.15           |
> | InternVL3    | 0.23                | 0.05           |
> | **Being-H**  | **0.58**            | **0.35**       |
> | **Being-H(FM)** | **0.63**         | **0.40**       |

---

> ### Author Response · Authors · 2025-11-28
> **Looking Forward to the Feedback on the Rebuttal Response**
>
> Dear Reviewer QutY,
>
> We hope this message finds you well. We sincerely appreciate your detailed feedback. In our earlier response, we provided concise clarifications on the points you raised including **clarity and notation, scalability, pipeline complexity, and the mathematical questions regarding our formulation**.
>
> As the rebuttal deadline approaches, we would like to kindly check whether our responses have resolved your concerns. If there are any additional questions or issues you would like us to clarify, please feel free to let us know. We will provide further feedback as soon as we can. We would be grateful if you could update your comments after reviewing our response.
>
> Warm regards,
>
> The Authors

---

### Official Review · Reviewer_dB8W · 2025-10-30

**Soundness:** 4
**Presentation:** 3
**Contribution:** 3
**Rating:** 6
**Confidence:** 5

**Summary:**

This paper proposes a paradigm that leverages large-scale human videos for VLA pretraining. The core method involves pretraining a model to generate detailed human hand motions from videos, and then adapting it to control a robot hand via post-training. The pretrained VLA successfully transfers human dexterity from internet videos to robots, demonstrating superior performance and data efficiency on real-world manipulation tasks compared to prior methods.

**Strengths:**

This paper's strength is its well-motivated idea to leverage human videos as a scalable resource for robot manipulation, a novel pretraining pipeline featuring part-level motion tokenization for high-fidelity control, and demonstrated strong performance with superior data efficiency on dexterous real-world tasks.

**Weaknesses:**

1. The approach is heavily reliant on the MANO hand model, which may not perfectly capture the full complexity and contact dynamics of real-world manipulation, potentially limiting the fidelity of the transferred skills. I suggest having some analytical experiments to assess the impact of potential errors in MANO on the entire pipeline.

2. The chosen tasks in the experiment section are not uniquely dependent on dexterous hands and could largely be accomplished with parallel grippers. The paper does not include functional grasping and in-hand manipulation tasks (e.g., reorienting a pen, spinning a key, or precise tool use) that would rigorously demonstrate the necessity of a dexterous hand and the specific advantages of the learned fine-grained finger control. This narrow task scope limits the claim of achieving general dexterous manipulation.

I would like to improve my score if the authors address my concerns during rebuttal.

**Questions:**

Please refer to the Weaknesses part.

---

> ### Author Response · Authors · 2025-11-22
> **Response to Reviewer dB8W**
>
> We thank the reviewer for the constructive feedback and for the positive assessment of the paper’s motivation, proposed tokenizer, and the demonstrated efficiency of transferring human motion priors. Regarding the reviewer's questions, we have updated a revised version with modified parts in dark red, and we respond to the raised concerns below.
>
> ### Impact of potential errors in MANO parameterization
> We thank the reviewer for raising this important point. We agree that MANO estimation is not perfect, but we posit that our pipeline is architecturally robust to such inaccuracies for two key reasons:
>
> 1. **Implicit Grounding Mitigates Retargeting Errors**
>
>     Our grounding is based on hidden states, which offers robustness without requiring explicit retargeting.
>     Specifically, our method does not perform explicit kinematic retargeting from human to robot joints. Instead, the pretrained VLA encodes behavior into abstract hidden states. The downstream robot-specific adapter then learns to map these states to feasible actions via imitation. Now we further strengthen this in Sec. 3.3.
>     This process naturally corrects for finger-level noise in the MANO estimates, as the robot is always grounded in its own embodiment.
>
> 2. MANO inaccuracies typically occur in fine-grained finger articulation, while the global trajectories and wrist motions are generally reliable. **During large-scale pretraining, such local inaccuracies act as small, unbiased noise**, and the model learns stable, generalized behavior priors rather than overfitting to potentially noisy per-frame finger details.
> This makes the learned priors robust and broadly transferable across tasks and embodiments.
> Resource constraints prevent us from running fully controlled “noisy-MANO pretraining” ablations.
>
> ### Tasks Requiring Dexterous Manipulation
>
> We thank the reviewer for raising this important point. In response, we added a new real-robot task that specifically requires dexterous, hand-like coordination far beyond simple pick-and-place:
> Spray-plant (one-handed grasp + trigger actuation)
> In this task, the robot must:
> 1. Grasp a spray bottle by its narrow neck.
> 2. Stabilize the spray bottle body using the thumb and the last three fingers.
> 3. Reserve and position the index finger to press the trigger.
> 4. Move the bottle to the plant and water it.
>
> This task requires:
> 1. precise grasp of affordance understanding,
> 2. differentiated finger roles (thumb/index vs. other fingers),
> 3. coordination between wrist alignment and finger actuation, and
> 4. continuous multi-contact stability.
>
> More details are provided in L352-356, L462-476, L1387-1394, and L1403-1408 of the revision. The results are:
> | Model        | Completion Score ↑ | Success Rate ↑ |
> |--------------|---------------------|----------------|
> | GR00T N1.5   | 0.33                | 0.15           |
> | InternVL3    | 0.23                | 0.05           |
> | **Being-H**  | **0.58**            | **0.35**       |
> | **Being-H(FM)** | **0.63**         | **0.40**       |
>
> Being-H and its variant (replacing MLP action head with a flow-matching one) achieves significantly higher success rates than both GR00T N1.5 and InternVL3, showing that the learned human-motion priors indeed provide dexterous coordination advantages.

---

> ### Author Response · Authors · 2025-11-28
> **Looking Forward to the Feedback on the Rebuttal Response**
>
> Dear Reviewer dB8W,
>
> We hope you are doing well and we sincerely appreciate your thoughtful feedback. In our earlier response, we provided concise clarifications on the points you raised including **MANO robustness, task dexterity, real robot performance, and the added dexterous spray plant task**.
>
> As the rebuttal deadline approaches, we would like to kindly ask whether our responses have resolved your concerns. If there are any additional questions or issues you would like us to clarify, please feel free to let us know. We would be grateful if you could update your comments after reviewing our response.
>
> Warm regards,
>
> The Authors

---

### Official Review · Reviewer_8ymu · 2025-10-31

**Soundness:** 3
**Presentation:** 3
**Contribution:** 3
**Rating:** 6
**Confidence:** 3

**Summary:**

This paper introduces Being-H, a large-scale vision-language-action (VLA) pretraining framework for dexterous robot manipulation.
The key idea is to treat human hand motion as a transferable prior for dexterous robot. Being-H first trains a transformer model on 2.5 M human video–text–motion triplets (the proposed UniHand-2.5M dataset) using MANO-based 3D hand parameters tokenized via a part-level grouped residual quantizer (GRQ). The pretrained model aligns vision, language, and hand motion tokens and is later adapted to robots through a lightweight projection and regression head for robot action prediction. Evaluations show that Being-H outperforms prior VLA baselines such as GR00T and InternVL3 on hand motion modeling, RoboCasa simulated tasks, and real-world dexterous manipulation, achieving higher success rates with only a fraction of teleoperation data.

**Strengths:**

1. Leveraging human-hand motion as pretraining data for dexterous robot control is a promising idea that directly targets the challenge of transferring human manipulation priors to robotic hands.

2. The work conducts large-scale multimodal pretraining on millions of human video–text–motion pairs and demonstrates clear empirical gains on hand motion modeling, simulated and real-world dexterous manipulation experiments.

3. The paper provides thorough experiments and analyses for the pretraining stage, including ablations on quantization design, model scale, and data efficiency, which make the technical contribution solid and well-validated.

**Weaknesses:**

1. The discussion of the embodiment gap between human and robot hands is not sufficiently clear. It remains unclear what specific aspects of human-hand pretraining help dexterous robot control, how large the embodiment gap actually is, and under what conditions the transfer succeeds or fails. The current framework behaves largely as a black box that relies on large-scale data to yield useful priors, without a mechanistic explanation.

2. The real-world evaluation appears incomplete and under-documented. For example, in Table 4, it is unclear how many trials were used to compute the success rate and how post-training performance scales with different amounts of robot data (e.g., 10, 50, or 100 demonstrations). It would also be important to compare against strong imitation-learning baselines such as Diffusion Policy, not just VLA-style models. Moreover, the paper does not provide any real-robot videos in the supplementary material, which makes it difficult to assess qualitative behavior or reproducibility.

**Questions:**

The paper appears to focus only on hand-centric representation learning and is applied to dexterous robotic hands. It would be helpful to clarify this scope explicitly in the title, so readers understand that the method is focusing on (dexterous) hand.

In Table 3, the performance on the Insertion task seems higher for GR00T than for Being-H.

---

> ### Author Response · Authors · 2025-11-22
> **Response to Reviewer 8ymu (1/2)**
>
> We thank the reviewer for the constructive feedback and for recognizing the novelty and value of leveraging large-scale human hand motion for dexterous VLA pretraining. Regarding the reviewer's questions, we have updated a revised version with modified parts in dark red, and we respond to the raised concerns below.
>
> ### Additional Discussion on human-to-robot transfer
> We thank the reviewer for pointing out this question and have added a detailed discussions in the revised manuscript, with the key points summarized as follows:
> 1. **Wrist and finger motion decomposition for embodiment tolerance**
>
>     Our method explicitly decomposes human hand motion into wrist and finger components.
>     - Wrist motions are highly morphology-invariant, capturing fundamental priors for end-effector approach, alignment, and trajectory that generalize across embodiments (DexHand, grippers, etc.).
>     - Finger motions are more embodiment-dependent. For human like hands (DexHand), they transfer directly. For grippers, they provide aggregated semantic and contact guidance (e.g., indicating where and how to make contact), without requiring kinematic replication.
>
>     This decomposition ensures that embodiment-specific contact mechanics do not corrupt the transferable global behavioral priors.
>
> 2. **Implicit Cross-Embodiment Mapping via Hidden States**
>
>     Instead of explicit kinematic retargeting, we bridge the embodiment gap through a learned mapping. Our pretrained VLA contributes rich behavior priors through its hidden states, where a lightweight MLP maps directly to the robot action space (Sec. 3.3). This implicit approach is unified and flexible, enabling it to support diverse downstream embodiments, even those with severe kinematic mismatch, such as grippers that cannot replicate human finger motions.
>
> 3. **Human priors provide consistent performance gains across a spectrum of embodiment gaps.**
>
>     The benefit of human-to-robot transfer scales with morphological similarity, yet remains significant even when the gap is large. Our experiments confirm this: while DexHand-based robots (smaller gap) exhibit the strongest transfer, enabling more complex tasks, gripper-like robots (larger gap) still show substantial gains, outperforming both InternVL3 and the robot-pretrained GR00T N1.5.
>     To further enlarge the evidence, we add:
>     - LIBERO experiments
>         | Model             | Spatial | Object | Goal | Long | Avg.  |
>         |-------------------|---------|--------|------|------|-------|
>         | Diffusion Policy  | 78.3    | 92.5   | 68.3 | 50.5 | 72.4  |
>         | Octo              | 78.9    | 85.7   | 84.6 | 51.1 | 75.1  |
>         | OpenVLA           | 84.7    | 88.4   | 79.2 | 53.7 | 76.5  |
>         | π₀-FAST           | **96.4**| **96.8**   | 88.6 | 60.2 | 85.5  |
>         | GR00T N1.5        | 92.0    | 86.0   | 92.0 | 76.0 | 86.5  |
>         | MolmoAct          | 87.0    | 95.4   | 87.6 | 77.2 | 86.6  |
>         | **Being-H**      | 92.6     | **96.8** | **94.4** | **77.4** | **90.3** |
>         | **Being-H(FM)**  | **95.2**   | **97.0** | **97.8** | **87.8** | **94.5** |
>     - a high-contact real-robot task: grasping and squeezing a spray bottle to water plants.
>        | Model        | Completion Score ↑ | Success Rate ↑ |
>         |--------------|---------------------|----------------|
>         | GR00T N1.5   | 0.33                | 0.15           |
>         | InternVL3    | 0.23                | 0.05           |
>         | **Being-H**  | **0.58**            | **0.35**       |
>         | **Being-H(FM)** | **0.63**         | **0.40**       |
>
>     Across all these embodiments, Being-H maintains a clear advantage. This is most pronounced in the spray-bottle task where human contact priors matter most, Being-H achieves a quite large margin of improvement.
> 4. **The capacity of the adaptation module directly influences the efficacy of transferred priors**
>
>     While our primary experiments use a lightweight MLP to successfully demonstrate human-to-robot transfer, we find that the choice of adapter is consequential. In extended experiments, replacing the MLP with a more expressive flow-matching head yielded further performance gains. This improvement is most pronounced for gripper-based robots, where the embodiment gap is largest.
>     This result confirms a key insight: while simple adaptation is sufficient for positive transfer, a more powerful mapping function is better equipped to bridge significant morphological differences, unlocking more of the potential held within the human priors.

---

> ### Author Response · Authors · 2025-11-22
> **Response to Reviewer 8ymu (2/2)**
>
> 5. **Future work**
>
>     We also acknowledge that our current approach, which relies solely on downstream adaptation, presents a limitation scope. A compelling direction for future work is to learn natively transferable priors by incorporating multi-embodiment robot data directly into pretraining. This could be achieved architecturally through multiple action heads or by constructing a unified action space shared across embodiments. The focus of this work, however, is to demonstrate that human videos with 3D hand motion alone constitute a powerful and highly scalable resource for VLA pretraining, a claim unequivocally supported by our extensive experimental results.
>
> ### Real-Robot Experimental Details
> Our original submission already used "20 randomized trials per task" (Sec 5.1) and "50–100 demonstrations for post-training" (Appendix E.2), which we now clarify further in the revision (25 per color for “Pick-Place-Toy”, 50 for other tasks). More details are provided in L340-358 and Appendix E.2.
> Figure 6 reports results in percentages to provide a consistent performance metric across tasks with varying absolute demonstration counts. All real-world experiment videos are now included in the Supplementary Materials.
>
> ### Post-training Data Scaling
> As shown in Figure 6, we have already investigated the impact of post-training data scale.
> To further strengthen this analysis, we now include results on full-view RoboCasa with 50, 100, and 200 demonstrations per task (totaling 1200, 2400, 4800 trajectories). More details are posted in Appendix F.4.
>
> | Demos (per task) | Pick & Place | Doors | Drawers | Levers | Knobs | Insertion | Buttons | Avg. |
> |------------------|--------------|-------|---------|--------|-------|-----------|---------|------|
> | 50×24 demos      | 10.0         | 64.5  | 48.0    | 58.0   | 21.0  | 26.0      | 34.0    | 33.5 |
> | 100×24 demos     | 16.0         | 71.5  | 55.0    | 60.7   | 25.0  | 27.0      | 38.0    | 38.5 |
> | 200×24 demos     | 18.0         | 76.0  | 63.0    | 66.7   | 24.0  | 31.0      | 41.3    | 42.0 |
>
> The results show a clear monotonic improvement, confirming that Being-H is not only effective in data-efficient, few-shot settings but also contributes to benefit from larger-scale robot data.
>
> ### Additional Baseline Comparisons
> Followed the reviewer’s suggestion, we report the performance of imitation learning baseline Diffusion Policy [1] from prior work in our new added LIBERO experiments.
> Being-H significantly outperforms Diffusion Policy, providing further evidence that pretraining on human motion offers a distinct advantage over standard visual imitation learning.
>
> ### Clarification on Hand-Centric Scope
> We thank the reviewer for this suggestion.
> To better reflect the paper's focus, we will update the title to explicitly include the term "Dexterous" in the final version ('Dexterous Vision-Language-Action Pretraining from Large-Scale  Human Videos'), clarifying that our proposed framework is designed for hand-centric VLA pretraining for dexterous manipulation.
>
>
> ### Correction to Table 3 and Pretraining Efficacy
> We mixed up the values for "insertion" and "buttons" in our previous submission. **We have fixed this typo in the updated version.**
> In the updated version, despite GR00T being pre-trained on extensive robot data, Being-H outperforms it on all the Robocasa tasks without any robot data pre-training.
> To better explain the effectiveness of our pre-training with human videos, we further report the performance of a tuned InternVL3 on RoboCasa as same as Being-H. Being-H shows a substantial improvement over InternVL3, reflecting a strong performance gain from our pretraining.
>
> | Model        | Pick&Place | Doors | Drawers | Levers | Knobs | Insertion | Buttons | Avg.  |
> |--------------|------------|-------|---------|--------|-------|-----------|---------|-------|
> | GR00T N1.5   | 1.3        | 40.5  | 37.0    | 48.0   | 11.0  | 6.0       | 26.7    | 21.0  |
> | InternVL3    | 1.3        | 37.0  | 36.0    | 42.0   | 9.0   | 4.0       | 18.0    | 18.2  |
> | **Being-H**  | **2.0**    | **43.5** | **40.0** | **51.3** | **11.0** | **17.0** | **30.7** | **23.8** |

---

> ### Author Response · Authors · 2025-11-28
> **Looking Forward to the Feedback on the Rebuttal Response**
>
> Dear Reviewer 8ymu,
>
> We hope this message finds you well. Thank you again for your insightful feedback on our work. In our earlier response, we provided concise clarifications on the points you raised including **embodiment gap, real robot details, data scaling, baseline comparisons, task specific corrections, and scope clarification**.
>
> As the rebuttal deadline approaches, we would like to kindly ask whether our responses have resolved your concerns. Please let us know if any points require further elaboration. We would be grateful if you could update your comments after reviewing our response, helping us to further refine our work!
>
> Warm regards,
>
> The Authors

---

### Official Review · Reviewer_qigd · 2025-11-01

**Soundness:** 3
**Presentation:** 4
**Contribution:** 3
**Rating:** 8
**Confidence:** 4

**Summary:**

This paper proposes a dexterous VLA training framework from large-scale human hand manipulation videos. It introduces a hand tokenization to tokenize the human hand for pretraining. Next, a post-training on the detextrous hand data is adapted. To support the training, a large-scale dataset is introduced. The experiments are conducted in different tasks.

**Strengths:**

1. The paper is well-written. The reviewer enjoys reading the introduction part of the paper, where some substantial challenges are discussed.
2. The whole training pipeline is well-designed, insightful and reasonable.
3. The dataset contribution is a great bonus, which is meaningful to the community.
4. The completeness of the paper is very good, with sufficient experiments (including real-world experiments) and supplementary material.

**Weaknesses:**

1. Based on the knowledge of the reviewer, there is still a large embodiment gap between human hands and robot manipulators, especially when the grasping ways or contact points are distinct between human and robot manipulators. In these extreme cases, does the proposed framework, especially the proposed hand motion representation/tokenizer, still work well?
2. Beyond the physical adaptation from the spatial perspective, is it possible to add a physical adaptation module considering the embodiement transfer (from human hand representation to robot manipulator representation) during post-training, to improve the better utilization of the pretrained priors?
3. Some failure case analysis and limitations should be discussed for future work.
4. The so-called "*physical space* alignment for 3D reasoning" or "physical tuning" is much overclaimed, as "physical" generally means a lot (not just depth or camera, also including reflection, force, material, dynamics, and so on). However, the proposed alignment is substantially a data distribution normalization/alignment for better pretraining. This should be addressed in the final version.
5. The ablation of post-training, such as scaling up the post-training data, using different post-training datasets (different gaps with pretraining dataset, also evaluate if can generalize to different post-training dataset) or networks, is not provided, which is similarly important for providing a clearer picture of the proposed framework.

**Questions:**

The reviewer expects some discussions about the questions raised in the weaknesses, and is glad to keep the rating if they are well-discussed.

---

> ### Author Response · Authors · 2025-11-22
> **Response to Reviewer qigd (1/2)**
>
> We sincerely thank the reviewer for thoughtful and constructive feedback, and we are so glad for the positive assessment of our paper’s writing quality, the overall training pipeline, the dataset contribution, and the completeness of the experiments. Regarding the reviewer's questions, we have updated a revised version with modified parts in dark red, and we respond to the raised concerns below.
>
> ### W1 & W2: Human-to-robot gap and physical adaptation considering the embodiement transfer
> Thank you for these insightful questions. They touch upon the core challenge of cross-embodiment transfer.
> Despite the challenge, we believe our framework can still works well and effectively leverage human videos to benefit robotic learning.
> Below we elaborate on the specific mechanisms that enable this adaptability, and provide additional analysis in order to maximize the benefit gained from human videos.
>
> 1. **Wrist-finger Decomposition Design and High-level Abstraction**
>
>     The wrist trajectory corresponds directly to the end-effector paths — a representation that is largely invariant across morphologies and is therefore highly transferable. To leverage this, our wrist-finger decomposition in part-level motion tokenizer (Sec. 3.1) separates globally-relevant trajectory (wrist motion) from embodiment-specific manipulation details (finger motion).
>     This decomposition ensures that embodiment-specific finger motions do not corrupt the transferable high-level behavior priors.
>
>     Furthermore, many manipulation contacts, the multi-finger hand can be modeled into simpler grippers. For example, thumb-finger opposition often approximates two-finger grasping. Our tokenizer leverages this by capturing contact-rich manipulation strategies at an abstract level that generalizes even when exact contact points differ between human hands and robot manipulators.
>
> 2. **We ground the pretrained human priors for downstream embodiments through imitation-oriented mapping.**
>
>     Instead of direct retargeting, our pretrained VLA provides abstract behavior priors via its hidden states. During downstream post-training and inference, an imitation objective maps these priors directly into the robot’s action space (Sec. 3.3). The process physically grounds the high-level human behavior (e.g., approaching patterns) into the robot's embodiment, enabling effective transfer even when low-level kinematics, like finger motions, are not reproducible.
>
> 3. **Performance might scale with morphological similarity to humans, but human priors provide a consistent advantage**
>
>     As expected, the performance gap may vary with embodiment similarity. Robots with human-like morphology (e.g., those based on the DexHand) exhibit a smaller gap, enabling richer transfer of complex skills. In contrast, simple grippers present a larger gap. Despite this, **Being-H outperforms all baselines on Robocasa, proving that human priors are beneficial even with significant morphological differences.** We further validate this by evaluating Being-H on the LIBERO benchmark (see L341-347, L408-425) and a contact-intensive real-world task (grasping and squeezing a spray bottle to water plants; see L352-356, L462-476, L1387-1394, L1403-1408). Across all settings, Being-H outperforms both the large-scale robot pretrained GR00T N1.5 and InternVL3 without any physical tuning. The advantage is most pronounced in the spray-bottle task, where nuanced contact priors are critical, demonstrating the clearest performance margin.
>
>
> 4. **The choice of adaptation modules influences the efficacy of the transferred human priors**
>
>     Our primary design uses a lightweight MLP projector for its directness and efficiency. In extended experiments, we also report a variant of Being-H, which replaces the MLP with a more expressive flow-matching action head, noted as Being-H (FM). We have elaborated this in Appendix B.4. **The results show that while the MLP successfully transfers high-level priors, the flow-matching head yields further performance gains.** This indicates that a more expressive action head can better bridge the physical gap between human priors and downstream policies when the embodiment discrepancy is large.
>
> 5. **Limitations and Further Plan**
>
>     A current limitation is that cross-embodiment transfer is achieved solely through post-training adaptation.
>     For future work, we plan to integrate multi-embodiment robot data directly into the pretraining phase.
>     This could be realized via multiple action heads (e.g., GR00T) or by constructing a unified action space shared across embodiments. The goal is to learn natively transferable priors for cross-embodiment, rather than being adapted solely afterward. The primary contribution of this work, however, is to robustly demonstrate that **human videos with hand motions alone provide a sufficiently rich and scalable source of behavior priors for VLA pretraining**, a claim consistently supported by our experimental results.

---

> ### Author Response · Authors · 2025-11-22
> **Response to Reviewer qigd (2/2)**
>
> **LIBERO Manipulation (Success Rate %)**
>
> | Model             | Spatial | Object | Goal | Long | Avg.  |
> |-------------------|---------|--------|------|------|-------|
> | Diffusion Policy  | 78.3    | 92.5   | 68.3 | 50.5 | 72.4  |
> | Octo              | 78.9    | 85.7   | 84.6 | 51.1 | 75.1  |
> | OpenVLA           | 84.7    | 88.4   | 79.2 | 53.7 | 76.5  |
> | π₀-FAST           | **96.4**| **96.8**   | 88.6 | 60.2 | 85.5  |
> | GR00T N1.5        | 92.0    | 86.0   | 92.0 | 76.0 | 86.5  |
> | MolmoAct          | 87.0    | 95.4   | 87.6 | 77.2 | 86.6  |
> | **Being-H**      | 92.6     | **96.8** | **94.4** | **77.4** | **90.3** |
> | **Being-H(FM)**  | **95.2**   | **97.0** | **97.8** | **87.8** | **94.5** |
>
> **Spray-Plant Dexterous Task**
>
> | Model        | Completion Score ↑ | Success Rate ↑ |
> |--------------|---------------------|----------------|
> | GR00T N1.5   | 0.33                | 0.15           |
> | InternVL3    | 0.23                | 0.05           |
> | **Being-H**  | **0.58**            | **0.35**       |
> | **Being-H(FM)** | **0.63**         | **0.40**       |
>
>
> ### W3: Failure cases and limitations.
> Thank you for pointing it out.
> We provide a detailed discussion in Appendix F.6. The primary limitations we observe arise mainly from **infrequent spatial perception deviations** and **minor inaccuracies in fine-grained control**. These issues largely stem from the depth ambiguity inherent in monocular RGB inputs and the difficulty of precisely grounding 3D motion from single-view observations. In tasks requiring subtle contact modulation, such inaccuracies may occasionally lead to failure even when the high-level strategy is correct.
>
> These findings motivate future work to incorporate multi-view inputs, explicit depth cues, and potentially more expressive control heads to further reduce perceptual ambiguity and enhance fine-grained dexterity.
>
> ### W4: “physical space alignment” and “physical tuning”.
> We thank the reviewer for this feedback.
> We will rename "physical space alignment" to the more precise "perspective spatial alignment", which specifically denotes unifying camera intrinsics and balancing viewpoint distribution to enable consistent 3D reasoning.
> However, "physical tuning" in our paper refers to the full pathway of grounding a VLM in the physical world, which consists of three levels:
> 	1.	Behavior grounding from human demonstrations: human motion tokens encode actionable physical priors beyond language semantics.
> 	2.	Perspective spatial alignment: unifying heterogeneous camera systems into a shared coordinate frame enables the model to reason in the real physical space beyond the ambiguous relative positions.
> 	3.	Downstream physical grounding: proprioception and robot-state feedback introduce dynamics, contact, and embodiment-specific physical signals unavailable in pretraining.
> While current pretraining captures spatial trajectories rather than low-level physical properties (e.g., force, compliance), we posit that spatial vision is the most scalable source of physical knowledge, while finer-grained physical signals can indeed be integrated during the downstream adaptation stages. We originally discussed this in Appendix C. Now we further clarify this hierarchy and terminology in the revision to avoid confusion.
>
> ### W5: Post-training ablation
> In our paper, we have presented an ablation on the demonstration scales used in the post-training(Fig. 6). In the extended experiments, we additionally evaluate the influence of different adaptation modules (Being-H v.s. Being-H (FM) in Tab. 4 and 6).
> Besides, we additionally post the scaling performance on RoboCasa (Details in Appendix F.4).
>
> | Demos (per task) | Pick & Place | Doors | Drawers | Levers | Knobs | Insertion | Buttons | Avg. |
> |------------------|--------------|-------|---------|--------|-------|-----------|---------|------|
> | 50×24 demos      | 10.0         | 64.5  | 48.0    | 58.0   | 21.0  | 26.0      | 34.0    | 33.5 |
> | 100×24 demos     | 16.0         | 71.5  | 55.0    | 60.7   | 25.0  | 27.0      | 38.0    | 38.5 |
> | 200×24 demos     | 18.0         | 76.0  | 63.0    | 66.7   | 24.0  | 31.0      | 41.3    | 42.0 |
>
> The ablation results show that Being-H benefits from human-motion pretraining and can work effectively even with a few demonstrations and a simple adaptation mechanism. And the performance scales along with the post-training demonstrations and adaptation module capacity.

---

> ### Author Response · Authors · 2025-11-28
> **Looking Forward to the Feedback on the Rebuttal Response**
>
> Dear Reviewer qigd,
>
> We hope this message finds you well. Thank you again for your insightful feedback on our work. In our earlier response, we provided concise clarifications on the points you raised, including the **embodiment gap, adaptation, failure cases, terminology, and post-training ablations**.
>
> As the rebuttal deadline approaches, we would like to kindly ask whether our responses have adequately addressed your concerns. Please let us know if any points require further elaboration. We would be truly grateful you find our revisions persuasive and could provide stronger comments in support of our work.
>
> Warm regards,
>
> The Authors

---

### Author Response · Authors · 2025-11-22
**General Response**

We sincerely thank all reviewers for their constructive feedbacks.

We are greatly encouraged by all reviewers' appreciation of this paper's insightful idea, promising implementaion and well-motivated approach (qigd, 8ymu, dB8W, QutY).
Additionally, the reviewers acknowledged the thorough experiments and analyses presented in the paper, including real-world dexterous experiments (qigd, 8ymu, QutY) as well as the resulting strong performance (8ymu, dB8W).
We are also pleased that our clear writing and the substantial challenges discussed in the introduction were recognized (qigd).

In response to the reviewers’ comments, we have updated a revised version of the submission. We add more clarifications of our methods for better logical flow and include additional experiments (covering LIBERO benchmark, better comparison on RoboCasa, a contact-intensive real-world task Spray-Plant，and scaling post-training on RoboCasa) to directly address reviewers’ concerns. These results consistently strengthen our conclusions and further validate the effectiveness of human-video-based VLA pretraining.

In the following sections, we provide detailed responses to the reviewers' comments, we would be more than happy to engage in further discuss with all reviewers!

---

### Meta-Review · Area_Chair_rcsh · 2026-01-07

**Summary:**

This work proposes a paradigm for incorporating large-scale human videos into VLA training and introduces MANO-based hand priors and discretized motion tokens during pretraining. The initial scores are 8662. The raised concerns include the cumulative errors from estimating MANO and how it can truly help difficult dextrous tasks, especially considering the human and robot hand. During rebuttal, the authors clarify that the learning is in the latent space and they add one dextrous task.

 My final decision is rejection considering:

1.  While significant effort is devoted to modeling hand motion at the pretraining stage, these explicit motion representations are discarded during post-training and are not used in robot control. As a result, human videos mainly contribute to semantic understanding and representation initialization, rather than directly improving manipulation behavior.  From this perspective, learning from latent action instead of explicit trajectories is not new in the field [1,2,3,4] . The authors fail to compare to them.


[1] UniVLA: Learning to act anywhere with task-centric latent actions. RSS25

[2] Latent action learning requires supervision in the presence of distractors. ICML25

[3] What Do Latent Action Models Actually Learn? NeurIPS25

[4] IGOR: Image-GOal Representations are the Atomic Control Units for Foundation Models in Embodied AI. ICLR25


2. Moreover, the method still relies heavily on expert demonstrations for robot control, and human videos do not fundamentally change the imitation-learning-based training paradigm. Consequently, the increased method complexity is not matched by proportional performance gains, and the potential of large-scale human videos for robot manipulation is not fully exploited.

3. Finally, though claiming to help with dextrous tasks, the evaluated tasks are overly simple, as metnioned by multiple reviewers, which significantly weaken the arguments of the work.

I encourage authors to submit to future venues with difficult dextrous tasks and comparisons with other latent action works.

**Reviewer Concerns:**

Most concerns are well-addressed. The remaining concerns regarding how the proposed method will actually help with true dextrous tasks beyond simple ones.

**Reviewer Scores:**

Reviewer QutY might raise the score to 4 since some of his concerns are solved. Other reviewers might maintain positive scores.

---

### Decision · Program_Chairs · 2026-01-26

Reject